# PertEval-scFM: Benchmarking Single-Cell Foundation Models for Perturbation Effect Prediction

## Abstract

*In silico* modeling of transcriptional responses to perturbations is crucial for advancing our understanding of cellular processes and disease mechanisms. We present PertEval-scFM, a standardized framework designed to evaluate models for perturbation effect prediction. We apply PertEval-scFM to benchmark zero-shot single-cell foundation model (scFM) embeddings against simpler baseline models to assess whether these contextualized representations enhance perturbation effect prediction. Our results show that scFM embeddings do not provide consistent improvements over baseline models, especially under distribution shift. Additionally, all models struggle with predicting strong or atypical perturbation effects. Overall, this study provides a systematic evaluation of zero-shot scFM embeddings for perturbation effect prediction, highlighting the challenges of this task and revealing the limitations of current-generation scFMs. Our findings underscore the need for specialized models and high-quality datasets that capture a broader range of cellular states. Source code and documentation can be found at: `https://anonymous.4open.science/r/PertEval-C674/`.

## 1 Introduction

Inspired by the success of foundation models in fields such as natural language processing (Devlin et al., 2019; Brown et al., 2020; OpenAI, 2024) and computer vision (Dosovitskiy et al., 2021), there has been an increase in the development of biological foundation models. Among these, single-cell foundation models (scFMs) leverage vast amounts of unlabeled transcriptomic single-cell RNA sequencing (scRNA-seq) data to learn contextualized representations through self-supervised pre-training (Ericsson et al., 2022). Fine-tuning the resulting model on labeled data enhances the performance on downstream applications, such as cell-type classification, gene regulatory network inference, and the prediction of cellular responses to perturbations (Yang et al., 2022; Kedzierska et al., 2023; Theodoris et al., 2023; Rosen et al., 2023; Cui et al., 2024; Wen et al., 2023; Hao et al., 2023).

A perturbation refers to any intervention or event leading to phenotypic alteration of a cell. Perturbation response prediction can provide invaluable insights into cellular mechanisms and disease progression, facilitating the mapping of genotype to phenotype and the identification of potential drug targets (Lotfollahi et al., 2019). Numerous models, here referred to as *narrow perturbation prediction models* (NPPMs), have been developed specifically for this task (Gavriilidis et al., 2024). However, perturbation response prediction is a challenging task, as demonstrated by the difficulty of models to improve consistently over simpler baseline methods (Wu et al., 2024; Branson et al., 2024; Ahlmann-Eltze et al., 2024).

Recently, there has been a concerted effort to evaluate biological foundation models. The Therapeutic Data Commons is an open science initiative that curates datasets, models and benchmarks related to a diverse range of therapeutic applications, including perturbation prediction (Velez-Arce et al., 2024). Additionally, Wu et al. (2024) and Ahlmann-Eltze et al. (2024) show that simple baseline models perform comparably to scFMs in predicting transcriptomic response to perturbations. However, their analysis does not account for distribution shift and focuses only on predictions for highly variable genes, many of which show little to no effect in response to a perturbation (Nadig et al., 2024).

Yet, distribution shift is a well-documented issue with scRNA-seq data (Boiarsky et al., 2023; Marklund et al., 2020). This often hinders the deployment of models that appear to perform well during evaluation. Distribution shift can occur as a consequence of inherent technical and biological noise, abundant in scRNA-seq data. While scFMs have been proposed to mitigate such problems, there have been conflicting reports on their ability to improve perturbation response prediction (Theodoris et al., 2023; Cui et al., 2024; Wu et al., 2024; Ahlmann-Eltze et al., 2024). This highlights the need for a comprehensive benchmark to evaluate their limitations and failure modes, specifically against distribution shift.

## 1.1 Contributions

Here, we present PertEval-scFM, a framework that addresses this research gap by providing:

- A detailed analysis of zero-shot scFM embeddings for perturbation effect prediction;
- A modular and extensible evaluation framework, with a toolbox of custom metrics designed to calculate and help interpret results;
- Integration of a spectral graph theory method – SPECTRA (Ektefaie et al., 2024) – that allows us to assess model generalizability under distribution shift.

We apply PertEval-scFM to investigate any added benefit of using scFM embeddings for perturbation response prediction. To do so, we use zero-shot embeddings generated from pre-trained scFMs and train an MLP probe (Jin et al., 2019). This allows for a fair evaluation of the transferability of these learned representations, without introducing inductive biases from different perturbation prediction models. The source code and documentation can be found on our GitHub.

## 2 PertEval-scFM Pipeline

In Figure 1 we present an overview of the PertEval-scFM pipeline, composed of three mains parts: data pre-processing, model training and evaluation. We define each part in the following section.

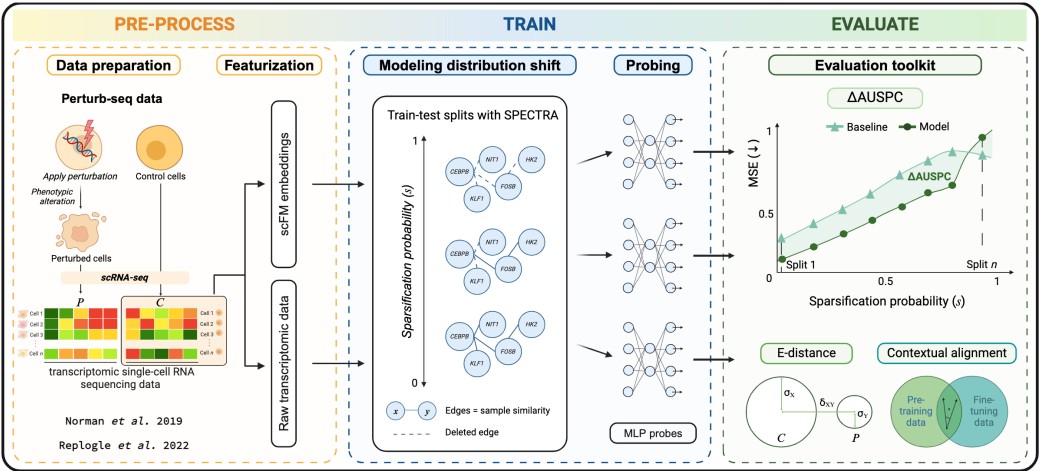

Figure 1: PertEval-scFM framework (left to right) – data pre-processing, training of MLP probes under different sparsification conditions; evaluation of trained models with AUSPC, E-distance and contextual alignment metrics.

## 2.1 Data pre-processing

To measure perturbation response we use Perturb-seq data, which integrates scRNA-seq with CRISPR-based perturbations to profile gene expression changes in response to specific genetic modifications

at the single-cell resolution (Dixit et al., 2016). Perturb-seq data consists of transcriptomic data for unperturbed control cells $C \in \mathbb{R}^{n_c \times g}$ and perturbed cells $P \in \mathbb{R}^{n_p \times g}$, where $n_c$ and $n_p$ corresponds to the number of control and perturbed cells being measured, and $g$ corresponds to the number of genes in the dataset.

### 2.1.1 DATA PREPARATION

PertEval-scFM takes as input the control cell matrix $C \in \mathbb{R}^{n_c \times g}$ obtained from Perturb-seq, containing the raw expression count. Briefly, our pre-processing pipeline consists of normalizing and log-transforming the raw expression count matrix. We then select the top 2,000 highly variable genes $v$ (HVGs), obtaining a reduced control matrix $C \in \mathbb{R}^{n_c \times v}$. We also calculate the differentially expressed genes (DEGs) for all perturbations to use in our evaluations. See Appendix A.2 for further details.

### 2.1.2 DATA FEATURIZATION

To generate the input features for our baselines, we randomly select 500 cells from $C$ to form a pseudo-bulk sample $\widetilde{C}$. To combat noise and sparsity issues, we calculate the average expression across $\widetilde{C}$ and repeat this process $n_p$ times. The resulting basal gene expression vectors can then be matched to perturbed cells, resulting in control expression feature matrix $X_c \in \mathbb{R}^{n_p \times v}$. See Appendix C.1 for further details.

**Single-cell foundation model embeddings.** To construct the control cell embeddings, we then feed our input matrix $X_c$ into the scFM:

$$f_{\text{scFM}}(X_c) = Z_c, \qquad Z_c \in \mathbb{R}^{n_p \times e} \tag{1}$$

where $e$ is the embedding dimension of the scFM. Perturbed cell embeddings $Z_p \in \mathbb{R}^{n_p \times e}$ are then generated by setting the expression of the perturbed genes to zero in all cells where it is expressed, effectively simulating a perturbation *in silico*. The control and perturbation embeddings are then concatenated to form the final input for the MLP probe. See Appendix C.2 for further details.

$$Z_{\text{scFM}} = Z_c \oplus Z_p \tag{2}$$

**Gene expression data embeddings.** To serve as a baseline against which to compare the performance of the scFM embeddings, we use our input matrix $X_c \in \mathbb{R}^{n_p \times v}$. Here, we model a genetic perturbation by calculating the gene co-expression matrix $G_c \in \mathbb{R}^{n_p \times v}$ between the perturbed genes and the highly variable genes in $X_c$. For two-gene perturbations, we calculate the co-expression matrices for each individual perturbation, and average the two to obtain $G_c$. We then concatenate the control and perturbation embeddings to form the final input for the MLP probe. See Appendix C.1 for further details.

$$Z_{\text{GE}} = X_c \oplus G_c \tag{3}$$

## 2.2 TRAINING

### 2.2.1 MLP PROBE FOR PERTURBATION EFFECT PREDICTION

A 1-hidden layer MLP was selected as a probe for its flexibility and simplicity in handling various types of data representations. For each perturbation, the MLP learns the log fold change perturbation effect $\delta$, defined as:

$$\delta := P - X_c \tag{4}$$

where $P \in \mathbb{R}^{n_p \times v}$ represents the perturbed gene expression matrix. The MLP probe predicts the perturbation effect, denoted by $\hat{\delta}$, described by the following equation:

$$\hat{\delta}^\theta(Z_{\text{scFM}}) = \text{ReLU}(Z_{\text{scFM}} W_1^\top + \mathbf{b}_1) W_2^\top + \mathbf{b}_2 \tag{5}$$

The model parameters $\theta$ include the weight matrices $W_1 \in \mathbb{R}^{h \times 2e}$ and $W_2 \in \mathbb{R}^{e \times h}$, where $h$ corresponds to the dimension of the hidden layer, and the bias vectors $\mathbf{b}_1 \in \mathbb{R}^h$ and $\mathbf{b}_2 \in \mathbb{R}^e$.

**MLP parameter count**. We train a range of MLPs with increasing parameter count on the log-normalized gene expression data to verify the effect of parameter count on results. We also include

additional results using scBERT and scFoundation embeddings as input, with increasing parameter count. We report our findings in Table D1, where it can be seen the increase in parameters has no effect on the MSEs obtained. Details on training and hyperparameter optimization are provided in Appendix D.2.

### 2.2.2 BASELINE MODELS

We establish baseline models against which to compare the performance of the MLP probes trained with scFM embeddings.

**Mean baseline.** The mean baseline assumes that a perturbation has little effect on the perturbed cell's gene expression. This reflects the biological reality that most perturbations result in small changes in gene expression, providing a simple biologically plausible null model highlighting the challenge inherent in distinguishing meaningful perturbation effects from background variability in single-cell data. The predicted perturbation effect, $\hat{\delta}$, is then simply computed as the deviation of the cell's gene expression, $X_c$, from the mean gene expression of all cells in the same context, $\overline{X}_c$, as defined by:

$$\hat{\delta} = \overline{X}_c - X_c \tag{6}$$

**MLP baseline.** The MLP baseline uses log-normalized gene expression data directly as an input. This approach ensures we can attribute any change in performance compared to the MLP baseline to the scFM embeddings.

$$\hat{\delta}^\eta(Z_{\text{GE}}) = \text{ReLU}(Z_{\text{GE}}W_1^\top + \mathbf{b}_1)W_2^\top + \mathbf{b}_2, \tag{7}$$

where dimensions of parameters $\eta$ correspond to $W_1 \in \mathbb{R}^{h \times 2v}$, $W_2 \in \mathbb{R}^{v \times h}$, $\mathbf{b}_1 \in \mathbb{R}^h$ and $\mathbf{b}_2 \in \mathbb{R}^v$.

**GEARS baseline.** GEARS is a state-of-the-art method for predicting perturbation effects on gene expression, integrating gene expression data with gene interaction networks through a graph-based framework (Roohani et al., 2023). We faithfully reproduced the original implementation, modifying only the train-test splits to align with the SPECTRA framework and evaluate robustness under distribution shift. All other training configurations, hyperparameters, and pre-processing steps followed the defaults provided in the GEARS implementation.

### 2.2.3 MODELING DISTRIBUTION SHIFT

To assess the robustness of the MLP probes when using either gene expression data or scFM embeddings, we implement SPECTRA (Ektefaie et al., 2024), a graph-based method that partitions data into increasingly challenging train-test splits while controlling for *cross-split overlap* between the train and test data.

In SPECTRA, edges within the graph represent sample-to-sample similarity. The connectivity of the similarity graph is controlled by the *sparsification probability* ($s$). For each split, this connectivity is adjusted by stochastically removing edges with sparsification probability $s \in [0, 1]$. We introduce the constraint $s < s_{\max}$, where $s_{\max}$ is empirically chosen to ensure a sufficient number of samples in both the train and test sets. After sparsification, the train and test sets are sampled from distinct subgraphs. As the sparsification probability increases, the degree of similarity between the train and test sets decreases, making it harder for the model to generalize to unseen perturbations effectively. For further details, see Appendix E.

### 2.3 EVALUATION

Currently, there is no consensus on how to benchmark perturbation effect prediction models. Here, we propose a standardized toolkit of three metrics, which aims to enhance model assessment, facilitate meaningful biological interpretation of results, and enable consistent cross-model comparisons:

- Area Under the SPECTRA Performance Curve (AUSPC)
- E-distance
- Contextual alignment

To assess model performance, we use the mean squared error (MSE) as our primary evaluation metric, based on prior work by Ji et al. (2023) demonstrating that the MSE provides a reliable assessment of perturbation effects reflective of biological reality.

### 2.3.1 AREA UNDER THE SPECTRA PERFORMANCE CURVE

To evaluate robustness under distribution shift, the AUSPC is adapted for perturbation effect prediction, following the approach introduced by Ektefaie et al. (2024). We formally define the AUSPC as:

$$\text{AUSPC} = \int_0^{s_{\max}} \phi(s)\,ds \tag{8}$$

where $\phi(s)$ is the MSE as a function of the sparsification probability $s$ used to define each train-test split. Integrating the MSE across $s$ yields a single performance metric that reflects a model's ability to generalize under increasing distribution shift. The integral is approximated with the trapezoidal rule (see Appendix E.2).

Motivated by the observation that simple baselines often perform surprisingly well in perturbation prediction, we introduce the $\Delta$AUSPC metric. This metric anchors a model's robustness to a baseline. The $\Delta$AUSPC is defined as:

$$\Delta\text{AUSPC} = \int_0^{s_{\max}} [\phi_b(s) - \phi_m(s)]ds \tag{9}$$

Here, $\phi_b$ represents the MSE of the mean expression baseline, and $\phi_m$ is the MSE of the model being evaluated. A positive $\Delta$AUSPC indicates that the model outperforms the baseline, while a negative value suggests the opposite. This metric provides a clear measure of a model's generalizability improvement over simply predicting the mean perturbation effect.

### 2.3.2 EVALUATING PERTURBATION STRENGTH USING E-DISTANCE

As introduced by Peidli et al. (2024), we use the E-distance as a metric to quantify the difference between perturbed and control cell gene expression profiles (Appendix F.1). This metric accounts for variability within and between the control and perturbed gene expression distributions, providing a quantitative measure of perturbation effect strength. This helps analyze the characteristics of perturbations that models succeed or struggle to predict accurately, helping to contextualize model performance, especially when dealing with outlier perturbations that traditional metrics may not immediately reveal.

### 2.3.3 CONTEXTUAL ALIGNMENT AND ITS EFFECT ON MODEL PERFORMANCE

While pre-training dataset size is often linked to improved downstream model performance, recent research emphasizes the critical role of data quality over dataset size (El-Nouby et al., 2021; Fournier et al., 2024). We therefore suggest the inclusion of a contextual alignment metric, which quantifies the similarity between the pre-training and fine-tuning datasets, and its effect on model performance. We calculate the cross-split overlap between the pre-train and fine-tune datasets using cosine similarity, to determine how representative the pre-training data is of the fine-tuning data (see Appendix G.1).

### 2.4 USE CASE

**Single-Cell Foundation Models.** PertEval-scFM currently includes the following scFMs: scBERT Yang et al. (2022), Geneformer (Theodoris et al., 2023), scGPT (Cui et al., 2024), scFoundation (Hao et al., 2023) and UCE (Rosen et al., 2023). In Table 1 we include details of their architecture and pre-training data. See Appendix B.1 for further details.

### 2.4.1 DATASETS

**Norman**. PertEval-scFM is applied to the 105 single-gene perturbations and 91 two-gene perturbations from the Norman et al. (2019) Perturb-seq dataset. This dataset contains high-quality CRISPRa perturbations in K562 cells, often used in perturbation prediction studies, as well as baseline expression for unperturbed cells. It allows for the systematic evaluation of model performance in predicting the effects of genetic perturbations at single-cell resolution.

Table 1: Overview of the scFMs included in PertEval-scFM.

| Model name | Architecture | Pre-training objective | # of cells | Organism | Emb. dim. |
|---|---|---|---|---|---|
| scBERT | Performer | Masked language modeling (MLM) | ∼5 million | human & mouse | 200 |
| Geneformer | Transformer | Masked language modeling (MLM) | ∼30 million | human | 256 |
| scGPT | Transformer | Specialized attention-masking mechanism | ∼33 million | human | 512 |
| UCE | Transformer | Masked language modeling (MLM) | ∼36 million | 8 species | 1,280 |
| scFoundation | Transformer | Read-depth-aware (RDA) modeling | ∼50 million | human | 3,072 |

**Replogle**. Additionally, we apply the framework to 1,866 single-gene perturbations from the Replogle et al. (2022) dataset, where CRISPRi has been used investigate knock-out transcriptomic perturbation response in K562 and RPE1 cells. In our work we focus on K562 cells in agreement with the Norman dataset. For details on the datasets, see Appendix A.

## 3 RESULTS

### 3.1 ZERO-SHOT SCFM EMBEDDINGS DO NOT MEANINGFULLY IMPROVE PERFORMANCE OVER RUDIMENTARY BASELINES ACROSS 2,000 HVGS

In Figure 2 and Table 2, we show that probes trained with zero-shot scFM embeddings did not show consistent improvement over the baseline models, with a 3.7% difference in AUSPC between Geneformer (worst) and the MLP baseline (best) for single-gene perturbations, and a 21.9% difference in AUSPC between scGPT (worst) and the MLP baseline (best). The performance metrics for single-gene perturbations showed no statistically significant differences between models, as evidenced by overlapping confidence intervals. In the case of two-gene perturbations, most models maintained comparable performance levels, while scGPT exhibited significantly lower performance. As the sparsification probabilities ($s$) increased from 0.0 to 0.7, the MSE worsened across all models. However, the zero-shot embeddings from the scFMs demonstrated a sharper decline in performance compared to the MLP baseline at higher sparsification probability values.

GEARS outperforms all zero-shot foundation models and baselines by an order of magnitude, suggesting that its architecture and training paradigm enable it to better capture the underlying biological processes and generalize more effectively across a wide range of perturbation scenarios. This superior performance highlights the necessity of strong inductive biases for gene perturbation prediction tasks, and suggests that representations that rely on a masked pre-training objective are only able to capture average perturbation effects at best. Overall, these results show that scFM embeddings do not mitigate problems caused by distribution shift and they do not provide a potent substrate to learn perturbation effects beyond average signal.

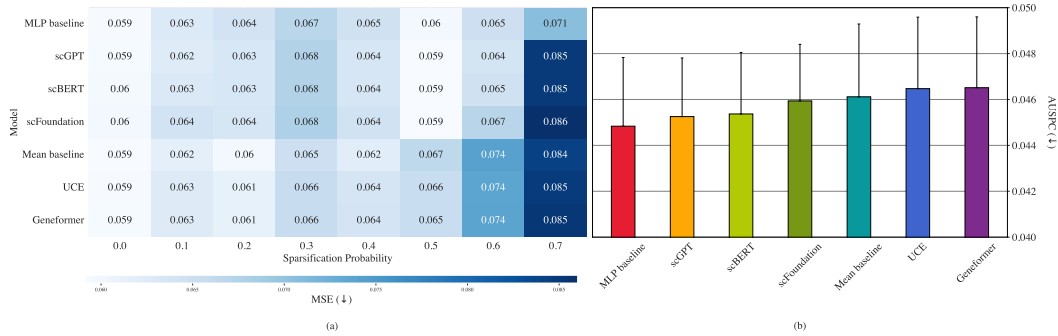

Figure 2: Perturbation effect predictions evaluated across 2,000 highly variable genes for 8 train-test splits of increasing difficulty. (a) MSE for all prediction models. Experiments were carried out in triplicate for each model. The heatmap shows the mean MSE values (↓). (b) Average AUSPC (↓) across sparsification probabilities for each model with standard error bars.

Table 2: Perturbation effect prediction evaluation across 2,000 HVGs. Models are listed in order of $\Delta$AUSPC. Asterisks (*) indicate that inference for these models is running. OOM: out of memory error (working on it).

| | | $\downarrow$ MSE $(10^{-2})$ | | | | | | | | $\downarrow$ AUSPC $(10^{-2})$ | $\uparrow \Delta$AUSPC $(10^{-2})$ |
|---|---|---|---|---|---|---|---|---|---|---|---|
| Model | Dataset | SP 0.0 | SP 0.1 | SP 0.2 | SP 0.3 | SP 0.4 | SP 0.5 | SP 0.6 | SP 0.7 | | |
| GEARS | | $0.550 \pm 0.023$ | $0.887 \pm 0.202$ | $0.937 \pm 0.177$ | $1.120 \pm 0.167$ | $1.693 \pm 0.328$ | $1.750 \pm 0.401$ | $1.0067 \pm 0.427$ | $1.000 \pm 0.257$ | $0.815 \pm 0.039$ | 3.7968 |
| MLP gene expression | | $5.935 \pm 0.213$ | $6.288 \pm 0.282$ | $6.410 \pm 0.289$ | $6.699 \pm 0.705$ | $6.453 \pm 0.584$ | $5.984 \pm 0.458$ | $6.502 \pm 1.277$ | $7.065 \pm 1.022$ | $4.484 \pm 0.299$ | 0.1280 |
| scGPT | | $5.940 \pm 0.207$ | $6.237 \pm 0.218$ | $6.340 \pm 0.608$ | $6.765 \pm 0.428$ | $6.363 \pm 0.345$ | $5.926 \pm 0.174$ | $6.400 \pm 1.144$ | $8.506 \pm 1.020$ | $4.525 \pm 0.255$ | 0.0863 |
| scBERT | Norman single-gene | $5.968 \pm 0.160$ | $6.301 \pm 0.316$ | $6.341 \pm 0.356$ | $6.761 \pm 0.765$ | $6.363 \pm 0.544$ | $5.924 \pm 0.418$ | $6.451 \pm 1.200$ | $8.488 \pm 0.558$ | $4.537 \pm 0.268$ | 0.0748 |
| scFoundation | | $5.989 \pm 0.162$ | $6.421 \pm 0.317$ | $6.366 \pm 0.356$ | $6.793 \pm 0.764$ | $6.440 \pm 0.538$ | $5.919 \pm 0.417$ | $6.705 \pm 1.183$ | $8.601 \pm 0.537$ | $4.594 \pm 0.246$ | 0.0179 |
| Mean baseline | | $5.916 \pm 0.161$ | $6.177 \pm 0.204$ | $5.980 \pm 0.621$ | $6.497 \pm 0.513$ | $6.219 \pm 0.308$ | $6.659 \pm 0.154$ | $7.413 \pm 1.038$ | $8.430 \pm 0.540$ | $4.612 \pm 0.317$ | - |
| UCE | | $5.937 \pm 0.140$ | $6.258 \pm 0.311$ | $6.132 \pm 0.620$ | $6.565 \pm 0.514$ | $6.387 \pm 0.307$ | $6.551 \pm 0.155$ | $7.370 \pm 1.065$ | $8.479 \pm 0.601$ | $4.647 \pm 0.312$ | -0.0355 |
| Geneformer | | $5.938 \pm 0.135$ | $6.257 \pm 0.049$ | $6.132 \pm 0.622$ | $6.565 \pm 0.520$ | $6.395 \pm 0.300$ | $6.550 \pm 0.140$ | $7.382 \pm 1.155$ | $8.525 \pm 0.494$ | $4.651 \pm 0.309$ | -0.0396 |
| GEARS | | $0.713 \pm 0.035$ | $0.783 \pm 0.044$ | $0.960 \pm 0.050$ | $1.153 \pm 0.049$ | $1.230 \pm 0.289$ | $1.467 \pm 0.351$ | $1.223 \pm 0.147$ | $1.810 \pm 0.287$ | $0.808 \pm 0.028$ | 0.043 |
| MLP gene expression | | $5.337 \pm 0.094$ | $5.261 \pm 0.100$ | $5.913 \pm 0.255$ | $5.728 \pm 0.402$ | $6.635 \pm 0.161$ | $7.675 \pm 0.953$ | $6.050 \pm 0.763$ | $5.198 \pm 0.593$ | $4.253 \pm 0.073$ | 0.002 |
| Mean baseline | | $5.337 \pm 0.093$ | $5.257 \pm 0.102$ | $5.910 \pm 0.255$ | $5.722 \pm 0.401$ | $6.644 \pm 0.167$ | $7.674 \pm 0.962$ | $6.071 \pm 0.772$ | $5.201 \pm 0.594$ | $4.255 \pm 0.073$ | - |
| scFoundation | Norman two-gene | $5.675 \pm 0.106$ | $5.564 \pm 0.051$ | $6.173 \pm 0.196$ | $6.050 \pm 0.462$ | $6.755 \pm 0.279$ | $7.944 \pm 1.186$ | $6.382 \pm 0.876$ | $5.238 \pm 0.578$ | $4.432 \pm 0.467$ | -0.177 |
| UCE | | $5.655 \pm 0.091$ | $5.514 \pm 0.066$ | $6.145 \pm 0.183$ | $6.029 \pm 0.460$ | $6.736 \pm 0.289$ | $7.939 \pm 1.184$ | $6.352 \pm 0.831$ | $5.612 \pm 0.665$ | $4.435 \pm 0.085$ | -0.180 |
| Geneformer | | $5.654 \pm 0.091$ | $5.514 \pm 0.067$ | $6.145 \pm 0.182$ | $6.029 \pm 0.458$ | $6.742 \pm 0.287$ | $7.937 \pm 1.187$ | $7.246 \pm 0.707$ | $5.179 \pm 0.630$ | $4.503 \pm 0.081$ | -0.248 |
| scBERT | | $5.655 \pm 0.092$ | $5.515 \pm 0.067$ | $6.159 \pm 0.196$ | $6.022 \pm 0.465$ | $6.757 \pm 0.281$ | $7.999 \pm 1.240$ | $6.493 \pm 0.736$ | $7.110 \pm 0.579$ | $4.533 \pm 0.081$ | -0.278 |
| scGPT | | $5.654 \pm 0.091$ | $5.515 \pm 0.067$ | $6.153 \pm 0.189$ | $6.023 \pm 0.464$ | $6.766 \pm 0.287$ | $8.272 \pm 1.377$ | $8.826 \pm 0.182$ | $14.906 \pm 2.154$ | $5.184 \pm 0.132$ | -0.929 |
| GEARS | | OOM | * | * | * | * | * | * | * | * | * |
| Geneformer | | OOM | 21.07 | 22.60 | * | * | 21.10 | * | * | * | * |
| Mean baseline | | * | 21.26 | * | * | * | * | * | * | * | * |
| MLP Baseline | Replogle | OOM | 21.12 | 22.70 | 22.08 | 21.54 | 21.20 | * | * | * | * |
| scBERT | | OOM | * | * | * | * | * | * | * | * | * |
| scFoundation | | OOM | 21.08 | 22.60 | 21.99 | 21.54 | 21.11 | * | * | * | * |
| scGPT | | OOM | 21.15 | 22.60 | 21.98 | 21.53 | 21.10 | * | * | * | * |
| UCE | | OOM | * | * | * | * | * | * | * | * | * |

## 3.2 Zero-shot scFM embeddings show minimal improvement over rudimentary baselines across the top 20 DEGs

Perturbations targeting few or even single genes typically alter the expression of a limited subset of genes within the transcriptome. Hence, models predicting mean gene expression can still achieve low MSE values across 2,000 HVGs. To better assess the ability of the models to predict meaningful perturbation effects, we restricted the evaluation to the top 20 DEGs per perturbation. The results are displayed in Table 3. This evaluation proves more challenging, evidenced by the order of magnitude increase in MSE (Appendix H.2). Consistent with the pattern observed for the 2,000 HVGs, the MSE values became worse as the sparsification probability increased, particularly for Geneformer and scGPT (Appendix H.3). For the single-gene perturbations, scBERT performed best across most sparsity levels ($\Delta$AUSPC = 0.00878), while UCE produced the most robust results ($\Delta$AUSPC = 0.0108). This indicates that these models were marginally better at capturing perturbation-specific expression changes in the top 20 DEGs, compared to the baselines. Conversely, Geneformer, scFoundation and scGPT showed negative $\Delta$AUSPC values, suggesting limitations in their ability to capture perturbation-specific expression changes. Despite these trends, the observed differences in performance were again minimal, with UCE (best) outperforming Geneformer (worst) by only 4.8%. These small differences and overlapping error margins suggest that no method provides significant performance gains over simpler approaches, even when focusing on the genes most affected by perturbations. The same pattern is observed for double-gene perturbations, where the models significantly outperform the mean baseline. However, consistent with our other results, the scFM embeddings still offer no advantage over the baseline MLP.

However, GEARS significantly outperforms all zero-shot foundation models and baselines (Table 3). For single-gene perturbations, GEARS achieves an AUSPC of 0.266, compared to 0.334 for UCE (the best among scFMs) and 0.342 for the MLP baseline, indicating a substantial improvement.

Table 3: Perturbation effect prediction evaluation across the top 20 DEGs per perturbation. Note that for double-gene perturbations split 0.5, there were not enough perturbations that passed our quality control to properly define the split.

| | | $\downarrow$ MSE | | | | | | | | $\downarrow$ AUSPC | $\uparrow \Delta$AUSPC $(10^{-2})$ |
|---|---|---|---|---|---|---|---|---|---|---|---|
| Model | Perturbation strategy | SP 0.0 | SP 0.1 | SP 0.2 | SP 0.3 | SP 0.4 | SP 0.5 | SP 0.6 | SP 0.7 | | |
| GEARS | | $0.240 \pm 0.025$ | $0.284 \pm 0.024$ | $0.215 \pm 0.037$ | $0.314 \pm 0.071$ | $0.256 \pm 0.035$ | $0.341 \pm 0.014$ | $0.682 \pm 0.194$ | $0.888 \pm 0.285$ | $0.266 \pm 0.018$ | 7.967 |
| UCE | | $0.355 \pm 0.037$ | $0.463 \pm 0.048$ | $0.464 \pm 0.077$ | $0.482 \pm 0.053$ | $0.476 \pm 0.042$ | $0.485 \pm 0.047$ | $0.484 \pm 0.104$ | $0.624 \pm 0.162$ | $0.334 \pm 0.012$ | 1.078 |
| scBERT | | $0.381 \pm 0.038$ | $0.469 \pm 0.050$ | $0.464 \pm 0.077$ | $0.481 \pm 0.053$ | $0.475 \pm 0.042$ | $0.482 \pm 0.045$ | $0.499 \pm 0.117$ | $0.608 \pm 0.149$ | $0.336 \pm 0.011$ | 0.878 |
| MLP gene expression | Single-gene | $0.379 \pm 0.038$ | $0.466 \pm 0.051$ | $0.468 \pm 0.074$ | $0.456 \pm 0.039$ | $0.497 \pm 0.042$ | $0.521 \pm 0.071$ | $0.513 \pm 0.123$ | $0.622 \pm 0.172$ | $0.342 \pm 0.013$ | 0.312 |
| Mean baseline | | $0.398 \pm 0.043$ | $0.479 \pm 0.050$ | $0.474 \pm 0.078$ | $0.489 \pm 0.053$ | $0.492 \pm 0.047$ | $0.492 \pm 0.047$ | $0.525 \pm 0.126$ | $0.604 \pm 0.144$ | $0.345 \pm 0.011$ | - |
| scGPT | | $0.402 \pm 0.035$ | $0.463 \pm 0.048$ | $0.464 \pm 0.077$ | $0.482 \pm 0.053$ | $0.475 \pm 0.042$ | $0.484 \pm 0.047$ | $0.485 \pm 0.105$ | $0.828 \pm 0.249$ | $0.347 \pm 0.015$ | -0.168 |
| scFoundation | | $0.406 \pm 0.041$ | $0.502 \pm 0.052$ | $0.466 \pm 0.077$ | $0.489 \pm 0.056$ | $0.469 \pm 0.040$ | $0.486 \pm 0.046$ | $0.567 \pm 0.090$ | $0.638 \pm 0.166$ | $0.350 \pm 0.011$ | -0.486 |
| Geneformer | | $0.405 \pm 0.044$ | $0.464 \pm 0.048$ | $0.464 \pm 0.077$ | $0.481 \pm 0.052$ | $0.475 \pm 0.042$ | $0.483 \pm 0.046$ | $0.488 \pm 0.106$ | $0.902 \pm 0.220$ | $0.351 \pm 0.014$ | -0.564 |
| GEARS | | $0.161 \pm 0.008$ | $0.211 \pm 0.032$ | $0.200 \pm 0.013$ | $0.296 \pm 0.052$ | $0.425 \pm 0.041$ | - | $0.473 \pm 0.109$ | $0.422 \pm 0.077$ | $0.223 \pm 0.010$ | 29.9 |
| MLP gene expression | | $0.195 \pm 0.121$ | $0.484 \pm 0.046$ | $0.538 \pm 0.082$ | $0.585 \pm 0.061$ | $0.618 \pm 0.048$ | - | $0.552 \pm 0.049$ | $0.500 \pm 0.056$ | $0.371 \pm 0.009$ | 15.1 |
| Geneformer | | $0.489 \pm 0.043$ | $0.527 \pm 0.055$ | $0.550 \pm 0.069$ | $0.603 \pm 0.076$ | $0.661 \pm 0.045$ | - | $0.623 \pm 0.054$ | $0.487 \pm 0.048$ | $0.409 \pm 0.008$ | 11.3 |
| UCE | Two-gene | $0.489 \pm 0.043$ | $0.527 \pm 0.055$ | $0.550 \pm 0.069$ | $0.601 \pm 0.072$ | $0.656 \pm 0.043$ | - | $0.624 \pm 0.053$ | $0.506 \pm 0.048$ | $0.410 \pm 0.007$ | 11.2 |
| scFoundation | | $0.493 \pm 0.044$ | $0.534 \pm 0.057$ | $0.554 \pm 0.070$ | $0.606 \pm 0.073$ | $0.656 \pm 0.045$ | - | $0.621 \pm 0.060$ | $0.497 \pm 0.051$ | $0.410 \pm 0.008$ | 11.2 |
| scBERT | | $0.490 \pm 0.043$ | $0.528 \pm 0.056$ | $0.550 \pm 0.069$ | $0.596 \pm 0.071$ | $0.661 \pm 0.041$ | - | $0.622 \pm 0.049$ | $0.681 \pm 0.068$ | $0.418 \pm 0.008$ | 10.4 |
| scGPT | | $0.489 \pm 0.043$ | $0.527 \pm 0.056$ | $0.550 \pm 0.069$ | $0.597 \pm 0.072$ | $0.673 \pm 0.044$ | - | $0.724 \pm 0.028$ | $1.941 \pm 0.329$ | $0.500 \pm 0.018$ | 2.2 |
| Mean baseline | | $2.524 \pm 1.054$ | $0.549 \pm 0.055$ | $0.580 \pm 0.075$ | $0.615 \pm 0.074$ | $0.653 \pm 0.037$ | - | $0.659 \pm 0.047$ | $0.497 \pm 0.056$ | $0.522 \pm 0.053$ | - |

The ΔAUSPC for GEARS is 7.967, markedly higher than the minimal gains observed for other models. This suggests that GEARS is better at capturing perturbation-specific expression changes, even when focusing on the genes most affected by perturbations. The performance gap widens for double-gene perturbations, where GEARS achieves an AUSPC of 0.223 and a ΔAUSPC of 29.9, outperforming the MLP baseline (AUSPC 0.371, ΔAUSPC 15.1) and all scFMs by a considerable margin. These results highlight the superior capability of GEARS to model complex gene interactions and perturbation effects, once again underscoring the importance of its architecture and training paradigm. In contrast, the zero-shot scFM embeddings offer no advantage over the baseline MLP, reinforcing our earlier conclusion that they do not provide significant performance gains, especially when focusing on the most affected genes.

### 3.3 E-DISTANCE ANALYSIS REVEAL FAILURE MODES OF PERTURBATION PREDICTION PROBES

Strong perturbation effects are generally under-represented in Perturb-seq data involving the perturbation of few genes. Hence, we hypothesized that models would struggle with predicting strong or atypically distributed perturbation effects. In Figure 3a we show the relationship between E-distance and performance, averaged across scFMs. Our E-distance analysis confirms that models generally perform worse when predicting the effect of perturbations with higher E-distance (i.e. strong perturbation effects). This trend was evident across all models, supporting the idea that training data with mild perturbation effects limits a model's ability to generalize to more extreme cases.

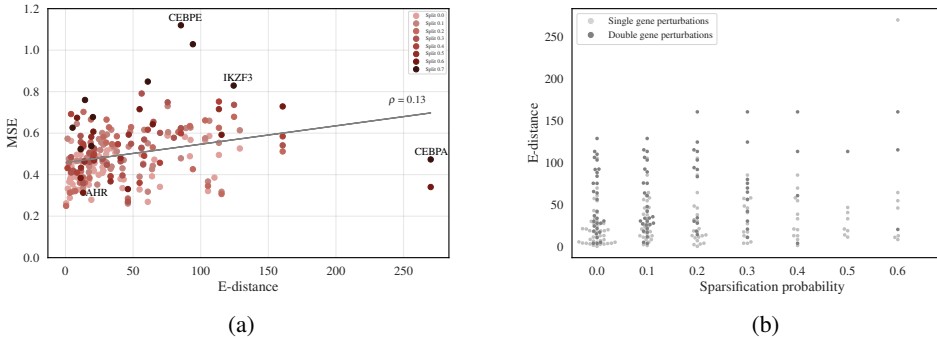

(a)                                                    (b)

Figure 3: (a) MSEs for all test perturbations as a function of the E-distance. The predictions displayed are the averaged across all scFMs. (b) The E-distance of all test perturbations stratified per split as a function of the sparsification probability. The mean of the E-distance per split is included in red.

Figure 3b further illustrates how perturbation strength is distributed across the different train-test splits for both single and double-gene perturbations. At higher sparsification probabilities, perturbations with lower E-distances become less frequent, while those with stronger effects appear more often. This is consistent with the earlier observation that performance declines as sparsity increases, as the models are increasingly challenged with stronger perturbations. Two perturbations illustrate this trend: *AHR* (Figure 4a), which has a low E-distance, showed a relatively small dynamic range in the target perturbation effects, ranging from about −0.1 to 0.25. In contrast, *CEBPE* (Figure 4b) showed a more pronounced perturbation effect, with a broader dynamic range of −0.5 to 1. The models performed worse when predicting the effect of *CEBPE* than that of *AHR*, aligning with our hypothesis that models poorly predict strong perturbations. This might be due to strong perturbations like *CEBPE* rarely appearing in the training data.

However, there are deviations from this trend. In Figure 4d, we show that *CEBPA*, which has a strong perturbation effect, was predicted relatively well by the models. Despite a high overall perturbation strength, *CEBPA* strongly modulates relatively few genes, with a longer tail of more mildly impacted genes. This suggests that the model's capability to predict perturbation effects depends not only on the magnitude of the perturbation, but also on its distribution. In Figure 4c, we show that *IKZF3* further substantiates this observation: despite eliciting a significantly weaker effect compared to

*CEBPA*, it was predicted less accurately, likely due to its atypical effect distribution (Appendix H.4). This suggests that model performance could be improved by more evenly representing perturbations across a wider range of effect sizes and distributions. These findings highlight the importance of exploring perturbation space more thoroughly and ensuring balanced representation during model training – a challenge that scFM embeddings alone are not equipped to address.

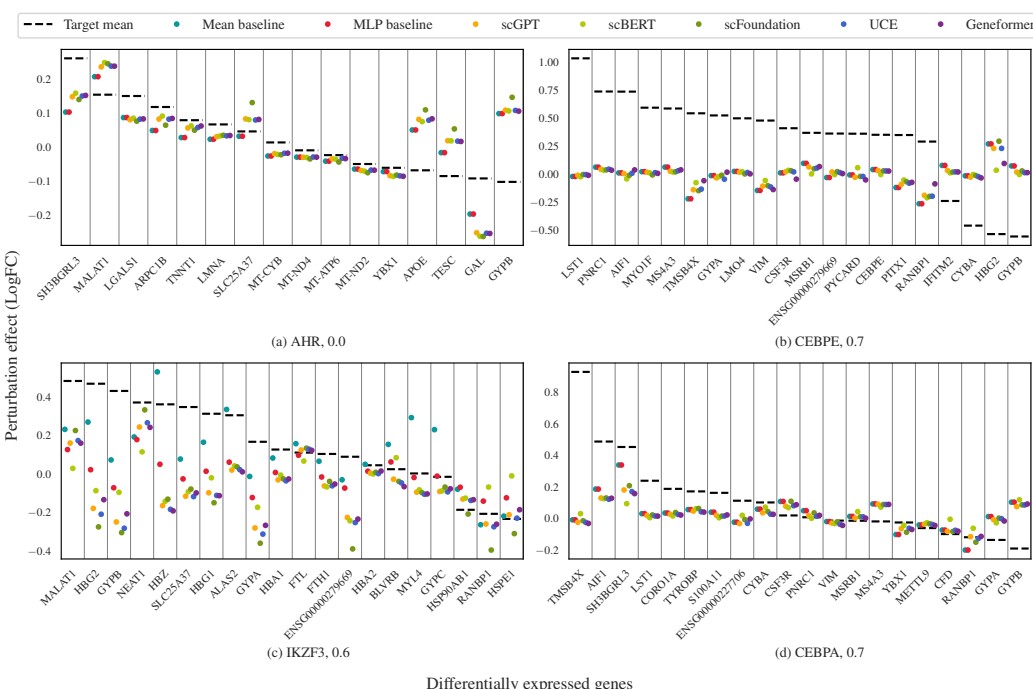

Figure 4: Predictions of models across the top 20 DEGs for 4 perturbations from different splits. Subcaptions indicate perturbation name, sparsification probability. The predictions are included as colored dots, and the target perturbation effect is displayed as a dashed line.

## 3.4 CONTEXTUAL ALIGNMENT BETWEEN PRE-TRAINING AND FINE-TUNING DATASETS HAS MINIMAL IMPACT ON INTRA-CELL TYPE PERTURBATION EFFECT PREDICTION

Previous research by Cui et al. (2024) demonstrated that the performance of models trained with zero-shot scFM embeddings is strongly affected by the overlap between their pre-training datasets and the downstream task data in cell-type annotation tasks. We sought to determine whether this reliance on contextual alignment extends to perturbation effect prediction.

In Figure 5, we calculated the contextual alignment between the datasets used to pre-train scGPT and scBERT, and the Norman dataset – used to fine-tune the scFM probes. The alignment scores were 0.606 and 0.718 for scGPT and scBERT, respectively, indicating that scBERT's pre-training corpus is approximately 19% more similar to the Norman dataset than that of scGPT. While the models show comparable MSE across splits, scBERT showed greater robustness. Notably, scGPT's pre-training corpus is an order of magnitude larger than scBERT's, underscoring the importance of contextual alignment over just scaling up the size of pre-training data.

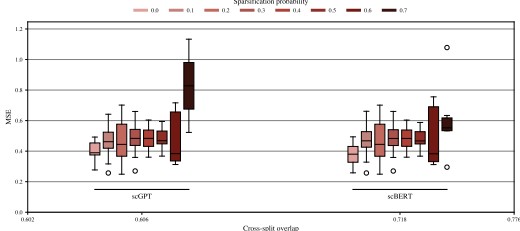

Figure 5: MSE as a function of the pre-train and fine-tune data cross-split overlap for scGPT and scBERT.

However, to fully appreciate the impact of contextual alignment in perturbation effect prediction, the experimental setup proposed here should be expanded to include a broader range of cell types and perturbations. We believe that future research should explore how contextual alignment may affect model performance when pre-training datasets are curated for perturbation effect prediction.

## 3.5 LIMITATIONS

In this study we focused on applying PertEval-scFM to one well-established high quality dataset. Ideally, we need to expand to more diverse datasets, including datasets containing chemical perturbations, to ensure the robustness of our framework and verify the findings presented here. Nonetheless, this is a step towards a unified framework to evaluate models for perturbation effect prediction.

## 4 CONCLUSION

PertEval-scFM addresses the current lack of consensus in benchmarking models for perturbation effect prediction by introducing a modular evaluation toolkit with diverse metrics designed to assess and interpret model performance. In particular, our framework allows consideration of distribution shift, often overlooked in other studies. We apply PertEval-scFM to evaluate the added benefit of using zero-shot scFM embeddings for perturbation prediction, instead of raw gene expression data. This study showed that current generation zero-shot scFM embeddings offer no improvement in perturbation effect prediction performance compared to rudimentary baselines when evaluated across 2,000 HVGs and 20 DEGs for single and double-gene perturbations. The AUSPC metric suggests that scFMs were less robust to distribution shift. Analysis using the E-distance metric revealed that the models particularly struggle to predict strong and atypically distributed perturbation effects. Finally, the contextual alignment metric points to the necessity of including a broader range of cell types and perturbations to better understand its impact on perturbation effect prediction. We plan to maintain and expand PertEval-scFM, developing a comprehensive benchmarking suite to facilitate the evaluation of perturbation models, and expect it to become a valuable community resource.

**Future work.** While our findings do not support the use of current-generation scFMs for reliable perturbation effect prediction, we recognize their potential. We expect that to make progress towards the accurate prediction of perturbation effects, scFMs must be customized for this task. Key questions, such as how to represent perturbations *in silico*, and how to fully leverage vast pre-training data, need to be addressed. Existing cell atlases only capture a tiny fraction of the human *phenoscape* – the full range of states possibly occupied by a cell (Fleck et al., 2023) – and often exclude perturbation-induced states. We think two key elements are required to improve the use of scFMs for perturbation effect prediction: higher-quality data that spans a wider range of the human phenoscape, covering multiple modalities, and consisting of clinically relevant cell types; and second, the development of specialized models, including scFMs, designed to fully leverage large-scale datasets to predict transcriptomic responses to perturbations – as exemplified by the superior performance of GEARS which includes inductive biases relevant to perturbation prediction.

## COMPUTATIONAL REQUIREMENTS

A single MLP probe was trained using 12 NVIDIA A100-PCIE-40GB GPU cores. Runtime depends on the hidden dimension of the probe, which is around 5 to 30 minutes for the smallest to biggest probes, respectively.

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

APPENDIX

# A  SINGLE-CELL DATA

The advent of single-cell RNA sequencing technology (scRNA-seq) has revolutionized our understanding of cellular heterogeneity and dynamic biological processes (Chen et al., 2019). Unlike traditional bulk sequencing methods, which average signals across large populations of cells, scRNA-seq technologies enable the study of gene expression at single-cell resolution. This granularity provides unprecedented insights into complex mechanisms of development, differentiation, and disease progression (Trapnell, 2015; Svensson et al., 2018; Fleck et al., 2023). The broad-scale application potential of scRNA-seq technology has led to the generation of large-scale datasets, such as the Human Cell Atlas (Regev et al., 2017) and the CellxGene Census (Program et al., 2023), which collectively span millions of cells and most sources of primary tissue.

## A.1  PERTURB-SEQ DATA

Perturb-seq integrates scRNA-seq with CRISPR-based perturbations to profile gene expression changes in response to specific genetic modifications at the single-cell resolution (Dixit et al., 2016). By systematically perturbing genes and measuring the resulting transcriptomic changes, Perturb-seq data provides a detailed map of cellular responses to specific genetic modifications. These datasets, such as those generated by Norman et al. (2019) and Replogle et al. (2022), allow researchers to explore the relationships between gene perturbations and cellular phenotypes in a high-dimensional space, providing invaluable insights into gene regulatory networks and cellular behavior and allowing the identification of potential drug targets (Wenteler et al., 2024).

### A.1.1  THE NORMAN DATASET

The dataset from Norman et al. (2019) represents one of the most comprehensive Perturb-seq resources available. It profiles transcriptional responses to over 100 single-gene perturbations in the human K562 leukemia cell line, using pooled CRISPR screening and scRNA-seq. This dataset captures gene expression data from thousands of individual cells, each subjected to either a control or a perturbation, providing an ideal testing ground for models designed to predict perturbation effects. The Norman dataset includes both perturbed and unperturbed cells, allowing for systematic evaluation of model performance in predicting the effects of genetic perturbations at single-cell resolution.

Table A1: Overview of the Norman dataset

| Characteristic | Description |
|---|---|
| Cell type | K562 (human leukemia cells) |
| Total number of perturbations | 196 |
| Number of single-gene perturbations | 105 |
| Perturbation method | CRISPRa |
| Number of control cells | ~12,000 |
| Number of cells | ~110,000 |
| Sequencing platform | 10x Genomics Chromium |
| Gene expression data | Single-cell RNA-seq |
| Number of genes measured | 20,000+ |
| Reference | Norman et al. (2019) |

## A.2 SINGLE-CELL DATA PRE-PROCESSING AND QUALITY CONTROL FUNCTIONS AND SETTINGS

The dataset was downloaded and pre-processed using `ScPerturb` (Peidli et al., 2024), `PertPy` (Heumos et al., 2024), and `ScanPy` (Wolf et al., 2018). As scFMs utilize raw gene expression counts, two versions of the dataset are stored internally: an `AnnData` object containing raw expression counts, used to generate embeddings with scFMs, and an `AnnData` object with pre-processed gene expression values, used to train the baseline models.

Pre-processing involved normalizing the raw gene expression counts by the total number of counts for each gene to account for differences in sequencing depth and ensure comparability across samples. This was performed using the `scanpy.pp.normalize_total(adata)` method with default settings. Next, the normalized counts were log-transformed with `scanpy.pp.log1p(adata)` to stabilize variance and make the data more amenable to downstream analysis. Finally, the top 2,000 highly variable genes were selected for training, using the `scanpy.pp.highly_variable_genes(pert_adata, n_top_genes=2000)` function.

## A.3 QUALITY CONTROL PLOTS

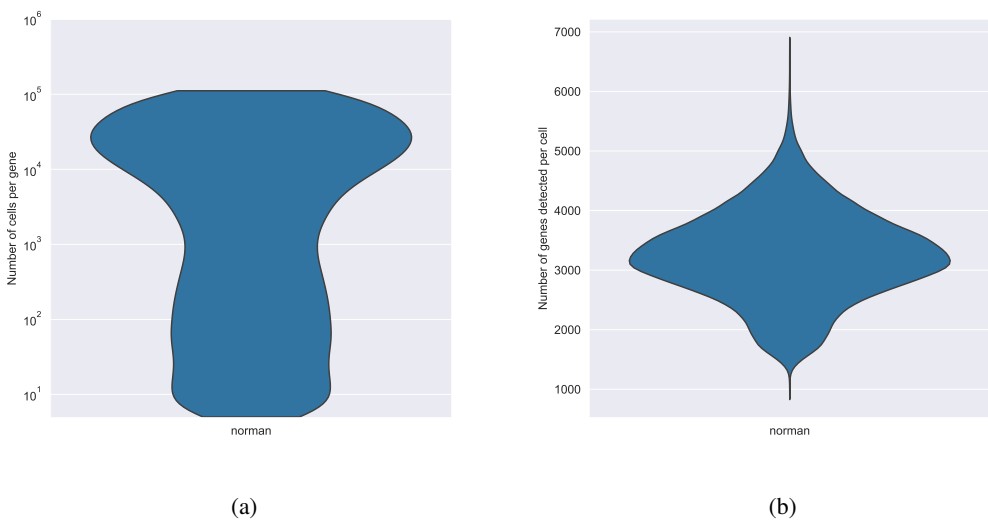

(a)                                                                 (b)

Figure A1: Quality control plots for the Norman dataset. (a) The number of cells per gene. This indicates how often an individual gene is measured across cells. Genes that are present in many cells might be housekeeping genes or essential genes. Because many genes were present in only a few cells, only genes present in minimum 5 cells were considered. (b) The number of genes detected per cell across all datasets. This offers insights into the distribution of genes among cells and indicates how representative the measurements are of single-cell transcriptomes.

# B  MODELS

## B.1  SINGLE-CELL FOUNDATION MODELS (SCFMS)

Single-cell foundation models (scFMs) are trained on broad single-cell data using large-scale self-supervision, allowing them to be adapted (i.e., fine-tuned) for a wide range of downstream tasks. Most scFMs use variants of the Transformer (Vaswani et al., 2017) architecture to process embedded representations of input gene expression data. However, they differ in input data representation, model architecture, and training procedures. Here, we provide a brief overview of the scFMs included in PertEval-scFM.

**Geneformer.**  Geneformer (Theodoris et al., 2023) employs six transformer units, each consisting of a self-attention layer and an MLP layer. The model is pre-trained on Genecorpus-30M, which comprises 29.9 million human single-cell transcriptomes from a broad range of tissues obtained from publicly available data. Before feeding the data into the model, gene expression values are converted into rank value encodings. This method provides a non-parametric representation of each single-cell transcriptome by ranking genes based on their expression levels in each cell and normalizing these ranks within the entire dataset. Consequently, housekeeping genes, which are ubiquitously highly expressed, are normalized to lower ranks, reducing their influence. Rank value encodings for each single-cell transcriptome are then tokenized, allowing genes to be stored as ranked tokens instead of their exact transcript values. Only genes detected within each cell are stored, thus reducing the sparsity of the data. When input into the model, genes from each single-cell transcriptome are embedded into a 256-dimensional space. Cell embeddings can also be generated by averaging the embeddings of each detected gene in the cell, resulting in a 256-dimensional embedding for each cell. The model is pre-trained using a masked learning objective, masking a portion of the genes and predicting the masked genes, which is intended to allow the model to learn gene network dynamics.

**scBERT.**  scBERT (Yang et al., 2022) adapts the BERT architecture (Devlin et al., 2019) for single-cell data analysis. A transformer is used as the model's backbone. The input data is represented as a sequence of gene expression values for each cell, where cells are constructed from gene expression value tokens. Gene embeddings are generated from the sum of two embeddings, where the first represents the gene's binned log-scale expression level, and the second is generated with gene2vec (Du et al., 2019) and specifies the gene's identity. The model is pre-trained via imputation on 5 million cells using a masked learning objective – masked gene expression values are predicted as a function of the other gene embeddings in the cell. In the paper, scBERT is fine-tuned for cell type annotation.

**scFoundation.**  scFoundation (Hao et al., 2023) employs xTrimogene as a backbone model, a scalable transformer-based architecture that includes an embedding module and an asymmetric encoder-decoder. The embedding module converts continuous gene expression scalars into high-dimensional vectors, allowing the model to fully retain the information from raw expression values, rather than discretizing them like other methods. The encoder is designed to only process nonzero and nonmasked gene expression embeddings, reducing computational load and thus enabling the application of *"vanilla transformer blocks to capture gene dependency without any kernel of low-rank approximation"*. These encoded embeddings are then recombined with the zero-expressed gene embeddings at the decoder stage to establish transcriptome-wide embedded representations. This backbone approach can then be built upon additional architectures which are specialized for specific tasks - i.e., GEARS (Roohani et al., 2023) for perturbation response prediction. scFoundation is pre-trained using read-depth-aware (RDA) modeling, an extension of masked language modeling developed to take the high variance in read depth of the data into account. The raw gene expression values are pre-processed using hierarchical Bayesian downsampling in order to generate the input vectors, which can either be the unchanged gene expression profile or where downsampling has resulted in a variant of the data with lower total gene expression counts. After gene expression has been normalized, raw and input gene expression count indicators are represented as tokens which are concatenated with the model input, allowing the model to learn relationships between cells with different read depths. Pre-training used data from over 50 million single cells sourced from a wide range of organs and tissues originating from both healthy and donors with a variety of diseases and cancer types.

**scGPT.** scGPT (Cui et al., 2024) follows a similar architectural and pre-training paradigm to scBERT. However, scGPT bins genes according to their expression, ensuring an even distribution across each bin. It uses random gene identity embeddings and incorporates an additional "condition embedding" to store meta-information and differentiate each gene. Along with gene embeddings, scGPT trains a cell token to summarize each cell. Instead of the long-range Performer architecture, scGPT processes embeddings via Flash-Attention (Dao et al., 2022) blocks. The model implements a generative masked pre-training using a causal masking strategy inspired by OpenAI's GPT series (Radford et al., 2018). scGPT is pre-trained on 33 million human cells and fine-tuned on a wide suite of downstream tasks, including cell type annotation, genetic perturbation response prediction, batch correction, and multi-omic integration.

**Universal Cell Embeddings (UCE).** Universal Cell Embeddings (UCE) (Rosen et al., 2023) is trained on a large compendium of single-cell RNA-seq datasets from multiple species, including human, mouse, mouse lemur, zebrafish, pig, rhesus macaque, crab-eating macaque, and western clawed frog, to create a universal embedding space for cells. The model converts the transcriptome of a single cell into an expression-weighted sample of its corresponding genes and then represents these genes by their protein products using a large protein language model. This representation is then fed into a transformer model. UCE is pre-trained in a self-supervised manner with a contrastive learning objective, where similar cells are mapped to nearby points in the embedding space, and dissimilar cells are mapped to distant points. This training paradigm enables UCE to provide high-quality embeddings that facilitate various downstream analyses. Benchmarks carried out by Rosen et al. (2023) in a zero-shot framework shown that UCE outperforms Geneformer (Theodoris et al., 2023) and scGPT (Cui et al., 2024), as well as cell annotation models such as scVI and scArches, in cell representation tasks.

### B.2 SCFM EMBEDDING GENERATION

In this section, we detail the process of generating embeddings for each foundation model in a zero-shot context using pre-trained models with frozen weights. For some models, pre-trained checkpoints are available and can be directly utilized, while others require initial pre-training. By freezing model weights, we ensure that the embeddings represent the learned features from the initial training phase, without further adaptation to the specific perturbation prediction task. This approach allows us to evaluate the inherent quality and utility of the pre-trained representations for downstream applications in biological research.

**Geneformer.** To generate embeddings for Geneformer (Theodoris et al., 2023), we downloaded the repository, including pre-trained model checkpoints, from Hugging Face. For control cells, we pre-processed the raw expression files to ensure the correct naming of columns and then fed them into the Geneformer tokenizer (`TranscriptomeTokenizer`). Once the dataset had been tokenized, we extracted embeddings using the pre-trained checkpoint (6-layer model) with the `EmbExtractor` method. For the perturbation data, we loaded the data and iterated through it in order to remove perturbed genes, simulating their deletion. The perturbed cells were then tokenized, and embeddings were extracted for each perturbed cell using the same functions.

**scBERT.** To generate emeddings for scBERT (Yang et al., 2022), we first downloaded the checkpoint and data shared in the scBERT GitHub repository. The environment was set up using the scBERT-reusability GitHub repository. For the raw expression counts, the genes were aligned using Ensembl *Homo sapiens* gene information. Log-normalization was performed and cells with less than 200 expressed genes were filtered out. For the perturbation data, the gene expression value was set to 0 to simulate perturbation, and embeddings were generated using the `predict.py` script.

**scFoundation.** To generate scFoundation embeddings (Hao et al., 2023), we initialized the scFoundation class shared at the official scFoundation GitHub repository. The `01B-resolution` pre-trained model checkpoint was loaded and the embeddings were generated while setting the `input_type = singlecell` and `tgthighres = f1` to indicate no read depth differences between unperturbed and perturbed cells. The embeddings were then generated using the `get_embeddings` function.

**scGPT.** To generate embeddings for scGPT (Cui et al., 2024) we installed the scGPT python package. We downloaded and used the whole-human scGPT model for embedding. For control cells, we used the scGPT `embed_data` function to generate the embeddings from the raw expression values. This function tokenises the data before feeding it through the model. For the perturbation data, we removed the perturbed genes, to simulate their deletion. The embeddings for the perturbed cells were then generated using the scGPT `embed_data` function.

**Universal Cell Embeddings (UCE).** To generate cell embeddings for UCE (Rosen et al., 2023), we ran the `eval_single_anndata.py` script provided in the UCE GitHub repository. Model weights for the 33-layer model and the pre-computed protein embeddings were downloaded separately from figshare. The script takes as input an h5ad raw expression file with variable names set as gene_symbols. The script was run with default parameters, except for the filter argument which was set to `False`, in order to skip an additional gene and cell filtering step. No further pre-processing was required to generate embeddings for control cells. For *in vitro* perturbed cells, the raw count value of the perturbed gene was explicitly set to zero for each condition prior to model inference, and saved as a h5ad file. The output of the script was an identical h5ad file with the input, except for cell-level embeddings that are stored in the `Anndata.obsm['X_uce']` slot.

## C    FEATURIZATION

### C.1    SINGLE-CELL EXPRESSION DATA FEATURIZATION

To generate the input features for raw single-cell expression data, we begin with the control matrix $C \in \mathbb{R}^{n_c \times v}$, consisting of $n_c$ unperturbed single-cell transcriptomes across $v$ highly variable genes (see Appendix A.2). From this matrix, we form a pseudo-bulk sample $\widetilde{C}$, which aggregates expression values from groups of cells within the same sample, in order to reduce sparsity and noise. Formally, let $\widetilde{C} = \{\mathbf{c}_i\}_{i=1}^{500}$ denote the set of randomly sampled cells from $C$. The average expression value $\overline{C}_j$ for each cell $j$ is then calculated by averaging the expression across the pseudo-bulked cells:

$$\overline{C}_j = \frac{1}{|\widetilde{C}|} \sum_{\mathbf{c}_i \in \widetilde{C}} c_{i,j} \quad \forall\, j \in \{1, \ldots, n_p\} \tag{C1}$$

Using this basal expression, we construct the input matrix $X_c \in \mathbb{R}^{n_p \times v}$, which has the same dimensions of the perturbed transcriptomic matrix $P \in \mathbb{R}^{n_p \times v}$ (i.e. what we want to predict), where $n_p$ is the number of perturbed cells. The input matrix $X_c$ is generated by sub-sampling from $\overline{C}_j$, ensuring that the dimensions are consistent between the input and the target output.

This approach ensures that input-target pairs are consistently defined for all training examples, as the dimensions of $X_c \in \mathbb{R}^{n_p \times v}$ align with the target matrix $P$. Representing input expression at pseudo-bulked basal levels helps mitigate sparsity issues caused by limited gene coverage in individual single-cell measurements from the original dataset. However, this method introduces a trade-off by reducing the heterogeneity of the input gene expression. As a result, some salient single-cell signals, such as those related to its initial state, may be diminished. However, inferring cellular states based solely on gene expression data is inherently challenging, given the many confounding factors and technical noise present in single-cell datasets (Fleming et al., 2023). Therefore, conventional machine learning models should not be expected to perform this task with high fidelity to begin with.

### C.1.1    MLP BASELINE

To generate the full set of input features for the MLP, we must encode the identity of each perturbation alongside capturing basal gene expression. Let $\mathcal{P} = \{p_1, \ldots, p_k\}$ denote the set of *perturbable* genes, and let $\mathcal{D} = \{d_1, \ldots, d_v\}$ represent all highly variable genes.

To evaluate the models' ability to generalize to unseen perturbations, it is important to incorporate information about gene interactions within a specific cell type. This allows the models to learn gene interaction networks, helping to extrapolate effects from known perturbations to novel ones.

To achieve this, we construct a $v$-dimensional correlation vector for each perturbable gene by calculating the Pearson correlation between its basal expression and that of all other genes, including itself. By including the auto-correlation of the perturbable gene, we explicitly encode the identity of the gene to be perturbed. The resulting feature vector for each perturbable gene, $\mathbf{g}_c \in \mathbb{R}^v$, captures the correlations between its basal expression and the basal expression of all highly variable genes. Aggregating these correlation vectors for all perturbable genes produces the matrix $G_c \in \mathbb{R}^{n_p \times v}$, where the perturbation in each row corresponds to the transcriptomic state observed in $T$.

Finally, the control gene expression matrix $X_c$ is concatenated with the perturbation correlation matrix $G_c$ to construct the complete input feature matrix:

$$Z_{\mathrm{GE}} = X_c \oplus G_c \tag{C2}$$

Here, $Z_{\mathrm{GE}} \in \mathbb{R}^{n \times 2v}$ represents the input feature matrix, where each row $\mathbf{g}_i$ combines the log-normalized basal expression values of a cell with the corresponding perturbation correlation features. This procedure is applied to both the training and testing sets, to generate $Z_{\mathrm{GE}_{\mathrm{train}}}$ and $Z_{\mathrm{GE}_{\mathrm{test}}}$.

## C.2 SINGLE-CELL FOUNDATION MODEL EMBEDDING FEATURIZATION

To generate embeddings from a pre-trained single-cell foundation model (scFM) with frozen weights, we begin by mapping raw gene expression counts to transcriptomic embeddings. Let $f_{\text{scFM}} : \mathbb{R}^l \to \mathbb{R}^{e_c}$ represent the function that transforms raw expression data into an embedding for each cell.

To construct the control cell embedding, we feed the raw expression vector $\mathbf{x}_i^c$ for each of the $n_c$ control cells into the scFM:

$$f_{\text{scFM}}(X_c) = Z_c \tag{C3}$$

The embedding vectors are then subsampled to create $\overline{Z}_c \in \mathbb{R}^{n_p \times e_c}$, where $n_p$ matches the number of perturbed cells and the dimension of $\overline{Z}_c$ aligns with the target output matrix.

An *in silico* perturbation embedding is then generated by nullifying the expression of the perturbed genes across all control cells in which it is expressed, up to a maximum of 500 cells. The nullification process, denoted by $N(\mathbf{x}_i^c, p_i)$, adjusts the gene expression vector according to the requirements of the scFM model in use. The nullification function can be defined as $N : \mathbb{R}^v \times \mathbb{N}_v \to \mathbb{R}^l$, where $\mathbb{R}^v$ represents the space of the gene expression vector, and $\mathbb{N}_v$ denotes the set of natural numbers from 1 to $v$, corresponding to the indices of genes in $\mathbf{x}_i^c$. If the scFM requires setting the perturbed gene's expression to zero, $l = v$. However, some scFMs filter out non-expressed genes during tokenization (scGPT), or train on ranked gene token representations instead of expression values (Geneformer). In these cases, the perturbed gene must be removed from the control gene expression vector, resulting in $l = v - 1$. Nonetheless, the perturbation embedding $\mathbf{x}_i^p$ is constructed as follows:

$$f_{\text{scFM}}(N(\mathbf{x}_i^c, p_i)) = \mathbf{z}_i^p \tag{C4}$$

The perturbation embeddings for all cells form the matrix $Z_p \in \mathbb{R}^{n_p \times e_c}$. It is trivial to extend the above framework to combinatorial perturbations, where the nullification function accepts multiple perturbations and nullifies the associated gene expression values.

The final cell embedding is then obtained by concatenating the control embedding $\overline{Z}_c$ with the perturbation embedding $Z_p$:

$$Z_{\text{scFM}} = \overline{Z}_c \oplus Z_p \tag{C5}$$

This approach differs from raw expression featurization, where co-expression patterns are explicitly encoded to model perturbations. In the scFM embedding featurization, *in silico* perturbation simulates the changes caused by gene perturbation. We hypothesize that the embeddings generated by scFMs inherently encode co-expression relationships, aligning with their pre-training objective based on masked language modeling.

In this study, zero-shot embeddings are generated using five different scFMs (Table 1). Inference for each scFM is tailored to the specific idiosyncrasies of the model in question. Detailed information on all the scFMs used can be found in Appendix B.1.

# D  MLP

## D.1  MLP PARAMATER COUNT

Table D1: Train and Test set results with MLPs of increasing parameter count

| Trainable parameters (million) | Training data | train/MSE | val/MSE |
|---|---|---|---|
| 1.6 | Raw gene expression | 0.057067 | 0.057642 |
| 3.2 | Raw gene expression | 0.058670 | 0.057493 |
| 6.3 | Raw gene expression | 0.056748 | 0.057424 |
| 12.7 | Raw gene expression | 0.056724 | 0.057428 |
| 1.6 | scFoundation embeddings | 0.060780 | 0.060260 |
| 3.2 | scFoundation embeddings | 0.060440 | 0.059910 |
| 12.6 | scFoundation embeddings | 0.059570 | 0.059050 |
| 0.2 | scBERT embeddings | 0.061040 | 0.061426 |
| 1.0 | scBERT embeddings | 0.061046 | 0.061428 |
| 8.0 | scBERT embeddings | 0.061040 | 0.061421 |

## D.2  HYPERPARAMETER OPTIMIZATION

To optimize the MLP probes, we used root mean square error (RMSE) as the objective function and the Adam optimizer (Kingma & Ba, 2017). Model performance was evaluated on an independent test set comprising unseen perturbations. The objective function to be minimized is:

$$\mathcal{L}(\theta) = \sqrt{\frac{1}{n_b} \sum_{j=1}^{n_b} \left( (T - X_c)_j - \hat{\delta}^\theta(X)_j \right)^2} \tag{D1}$$

where $j$ indexes each cell and $n_b$ denotes the batch size.

Hyperparameters were selected using the tree-structured Parzen estimator (TPE) tuning algorithm (Bergstra et al., 2011). This optimization was performed on the first train-test split, which contains the largest training set. Given the computational demands of exhaustive parameter sweeps, we focused on optimizing the hyperparameters using the gene expression data as a reference.

An initial search across different numbers of hidden layers revealed that this parameter had no substantial effect on model performance. Therefore, a single hidden layer was used throughout the experiments to maintain model simplicity. The learning rate, however, was found to significantly influence performance and was thus adjusted for the models trained using the scFM embeddings. Following the manifold hypothesis, we set the hidden dimension to half of the input dimension (Bengio et al., 2013). A comprehensive list of the final hyperparameters for each model is provided in Table D2.

Table D2: TPE hyperparameter optimization results for all the datasets and probes considered.

| Dataset | Model Type | Hyperparameter | | |
|---------|-----------|----------------|---|---|
| | | **Type** | **Name** | **Value** |
| Norman | MLP Gene expression | Adam Optimizer | Starting Learning Rate | $5 \cdot 10^{-5}$ |
| | | | Max. Epochs | 100 |
| | | ReduceLROnPlateau Scheduler | Reduction Factor | 0.1 |
| | | | Patience | 15 |
| | | | Threshold | $1 \cdot 10^{-4}$ |
| | | | Min. Learning Rate | $5 \cdot 10^{-9}$ |
| | | Model | Hidden Layers | 1 |
| | | | Hidden Dimension | $1,024$ |
| | | Data | Batch Size | 64 |
| Norman | MLP Geneformer | Adam Optimizer | Starting Learning Rate | $3 \cdot 10^{-4}$ |
| | | | Max. Epochs | 100 |
| | | ReduceLROnPlateau Scheduler | Reduction Factor | 0.1 |
| | | | Patience | 10 |
| | | | Threshold | $1 \cdot 10^{-4}$ |
| | | | Min. Learning Rate | $5 \cdot 10^{-9}$ |
| | | Model | Hidden Layers | 1 |
| | | | Hidden Dimension | 128 |
| | | Data | Batch Size | 64 |

Table D3: TPE hyperparameter optimization results, continued.

| Dataset | Model Type | Hyperparameter | | |
| | | Type | Name | Value |
|---|---|---|---|---|
| Norman | MLP scBERT | Adam Optimizer | Starting Learning Rate
Max. Epochs | $5 \cdot 10^{-6}$
100 |
| | | ReduceLROnPlateau Scheduler | Reduction Factor
Patience
Threshold
Min. Learning Rate | 0.1
10
$1 \cdot 10^{-4}$
$5 \cdot 10^{-9}$ |
| | | Model | Hidden Layers
Hidden Dimension | 1
100 |
| | | Data | Batch Size | 64 |
| Norman | MLP scGPT | Adam Optimizer | Starting Learning Rate
Max. Epochs | $3 \cdot 10^{-4}$
100 |
| | | ReduceLROnPlateau Scheduler | Reduction Factor
Patience
Threshold
Min. Learning Rate | 0.1
10
$1 \cdot 10^{-4}$
$5 \cdot 10^{-9}$ |
| | | Model | Hidden Layers
Hidden Dimension | 1
256 |
| | | Data | Batch Size | 64 |
| Norman | MLP UCE | Adam Optimizer | Starting Learning Rate
Max. Epochs | $3 \cdot 10^{-4}$
100 |
| | | ReduceLROnPlateau Scheduler | Reduction Factor
Patience
Threshold
Min. Learning Rate | 0.1
10
$1 \cdot 10^{-4}$
$5 \cdot 10^{-9}$ |
| | | Model | Hidden Layers
Hidden Dimension | 1
640 |
| | | Data | Batch Size | 64 |
| Norman | MLP scFoundation | Adam Optimizer | Starting Learning Rate
Max. Epochs | $3 \cdot 10^{-4}$
100 |
| | | ReduceLROnPlateau Scheduler | Reduction Factor
Patience
Threshold
Min. Learning Rate | 0.1
10
$1 \cdot 10^{-4}$
$5 \cdot 10^{-9}$ |
| | | Model | Hidden Layers
Hidden Dimension | 1
$1,536$ |
| | | Data | Batch Size | 64 |

# E SPECTRA

## E.1 EVALUATING MODEL ROBUSTNESS UNDER DISTRIBUTION SHIFT IN SINGLE-CELL DATA WITH SPECTRA

Sample-to-sample similarity must be calculated to construct the spectral graph for single-cell data. If two samples are sufficiently similar, an edge will be inserted in the spectral graph. To quantify sample-to-sample similarity between distributions, the L2 norm, denoted by $\| \cdot \|$, of the $\log 1p$-fold change between the mean perturbation expression vector, $\overline{\mathbf{p}}_i$, and the mean control gene expression vector, $\overline{\mathbf{c}}$, is calculated:

$$S(\overline{\mathbf{p}}_i, \overline{\mathbf{c}}) = \| \log(\overline{\mathbf{p}}_i + 1) - \log(\overline{\mathbf{c}} + 1) \| \tag{E1}$$

Using this definition, a series of train-test splits are generated by sparsifying the initial graph. Train and test instances are samples from distinct subgraphs for each split, with decreasing mean pairwise similarity between the two sets. The sparsification of the initial graph is attenuated by a *sparsification probability* ($s$), which is the probability that an edge between two samples will be be dropped. Mathematically, SPECTRA employs a graph sparsification technique similar to what is described in Spielman & Teng (2010). A practical limitation of the current implementation of SPECTRA lies in its tendency to unevenly distribute perturbations of similar magnitudes across the training and test splits while minimizing cross-split overlap. This uneven distribution engenders class imbalances that become increasingly pronounced at higher sparsification probabilities. Consequently, this imposes a trade-off between induced class imbalance and simulated distribution shift. Empirical observations on the Norman data indicate that the sparsification probability threshold at which the class imbalance remains manageable is approximately 0.7. Beyond this threshold, the deleterious effects of class imbalance as well as low sample numbers begin to outweigh the benefits of reduced cross-split overlap.

For the Norman dataset, Appendix E1a illustrates a rapid decrease in the number of training and testing samples as the sparsification probability increases. This is expected, as a higher sparsification probability leads to increasingly disconnected subgraphs to draw samples from. Furthermore, appendix E1 confirms that SPECTRA can simulate distribution shift by showing a corresponding decrease in similarity between the samples as sparsification probability rises. Subsequently, we train and test models on each SPECTRA split and plot the MSE as a function of the decreasing cross-split overlap. The area under this curve is defined as the AUSPC, which serves as a measure of model generalizability under distribution shift.

Similarly to the within-dataset case outlined above, the cross-split overlap can be used to measure the similarity between-datasets, in this case between the scFM pre-train and our fine-tune datasets for scBERT and scGPT. This approach allows us to investigate the impact of pre-training data on the quality of scFM embeddings. Further details are provided in Section G.1.

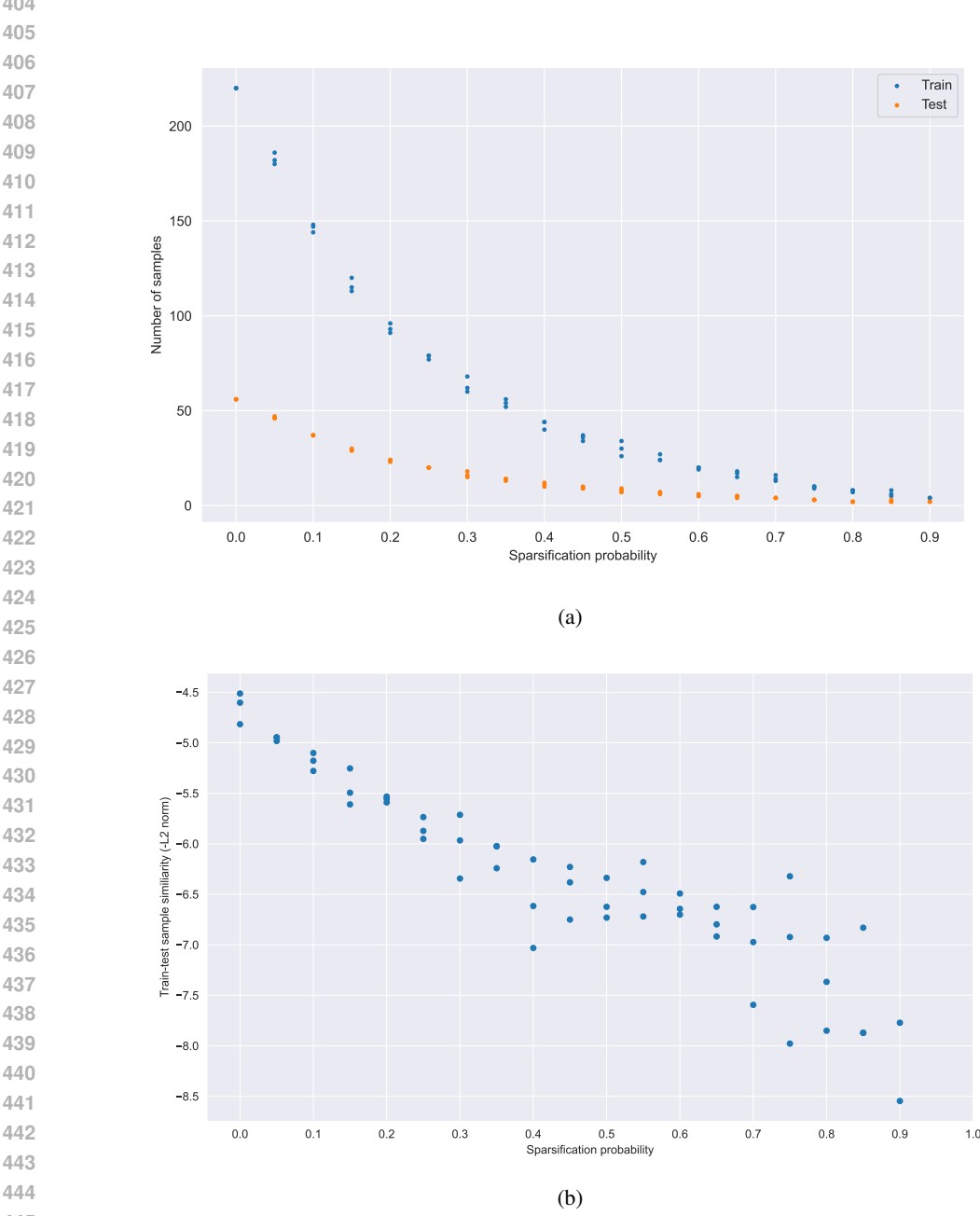

(a)

(b)

Figure E1: (a) Number of samples in train and test as a function of the sparsification probability. (b) Cross-split overlap as a function of the sparsification probability.

### E.2 IMPLEMENTATION DETAILS OF THE AUSPC

The AUSPC is defined by Equation 8. For numerical evaluation, the integral is approximated using the trapezoidal rule with sparsification probabilities $s_i \in \{0.0, 0.1, ..., 0.7\}$:

$$\begin{aligned} \text{AUSPC} = f(\phi) &= \int_0^{s_{\max}} \phi(s)\, ds \\ &\approx \frac{d}{2} \sum_{i=0}^{n-1} [\phi(s_i) + \phi(s_{i+1})] \end{aligned} \tag{E2}$$

where $d$ denotes the step size of the sparsification probability (0.1 in this case) and $\phi$ represents the metric of interest, (MSE). The $\Delta$AUSPC is subsequently derived by calculating this value for both the baseline and the model independently, and then subtracting the AUSPC of the model from that of the baseline. For simplicity, we use the notation $\phi_i = \phi(s_i)$.

To quantify the uncertainty associated with the AUSPC, uncertainty propagation is utilised, wherein the AUSPC is assumed to be a non-linear function of the metric of interest, $\phi(s)$. For uncertainty propagation in this context, the following equation is employed:

$$\sigma^2 = \sum_{i=1}^{n-1} \left( \frac{\partial f}{\partial \phi_i} \sigma_{\phi_i} \right)^2 \tag{E3}$$

where $\sigma$ represents the total error associated with the AUSPC and $\sigma_{\phi_i}$ denotes the error associated with the MSE for split $i$.

The partial derivative $\frac{\partial f}{\partial \phi_i}$ is calculated using the definition of $f$ given in Equation E2:

$$\begin{aligned} \frac{\partial f}{\partial \phi_i} &= \frac{d}{2} \sum_i \left( \frac{\partial}{\partial \phi_i} \phi_i + \frac{\partial}{\partial \phi_i} \phi_{i+1} \right) \\ \frac{\partial f}{\partial \phi_i} &= \frac{d}{2} \end{aligned} \tag{E4}$$

Substituting this result into Equation E3 yields:

$$\begin{aligned} \sigma^2 &= \sum_i \left( \frac{d}{2} \sigma_{\phi_i} \right)^2 \\ \sigma &= \sqrt{ \sum_i \left( \frac{d}{2} \sigma_{\phi_i} \right)^2 } \end{aligned} \tag{E5}$$

The algorithmic implementation is given in Algorithm 1.

---

**Algorithm 1** Calculate AUSPC and its associated error

---

1: **function** TRAPEZOIDALAUSPC($\phi$, $s$)
2:     AUSPC $\leftarrow$ `np.trapz`($\phi$, $s$)
3:     **return** AUSPC
4: **end function**
5: **function** CALCULATEDELTAAUSPC($\phi_b$, $\phi_m$, $\sigma_b$, $\sigma_m$, $s$)
6:     AUSPC$_b$ $\leftarrow$ TRAPEZOIDALAUSPC($\phi_b$, $s$)
7:     AUSPC$_m$ $\leftarrow$ TRAPEZOIDALAUSPC($\phi_m$, $s$)
8:     $d \leftarrow s[1] - s[0]$                                   $\triangleright$ Assuming uniform step size
9:     $\sigma_b \leftarrow \sqrt{\sum_i (\frac{d}{2}\sigma_{\phi_b,i})^2}$
10:     $\sigma_m \leftarrow \sqrt{\sum_i (\frac{d}{2}\sigma_{\phi_m,i})^2}$
11:     $\Delta$AUSPC $\leftarrow$ AUSPC$_b$ $-$ AUSPC$_m$
12:     **return** $\Delta$AUSPC, $\sigma_b$, $\sigma_m$
13: **end function**

---

# F    E-STATISTICS

## F.1    USING E-DISTANCE AND DIFFERENTIAL GENE EXPRESSION TO EVALUATE SIGNIFICANT PERTURBATIONS

While examining transcriptome-wide, aggregated perturbation effects provides valuable insights, it lacks the granularity needed to assess a model's ability to reconstruct perturbation effects at the gene level. To address this limitation, energy statistics (E-statistics) are employed to evaluate and select significant perturbations in single-cell expression profiles. Subsequently, differential gene expression analysis is carried out to identify the top 20 differentially expressed genes which are then used to evaluate individual perturbations.

Perturbation effects are quantified using the E-distance, which compares mean pairwise distances between perturbed and control cells. Let $\mathcal{X} \in \{\mathbf{x_1}, \ldots, \mathbf{x}_{n_a}\}$ and $\mathcal{Y} \in \{\mathbf{y_1}, \ldots, \mathbf{y}_{n_b}\}$ be two distributions of cells in different conditions with $n_a$ and $n_b$ cells respectively, where $\mathbf{x}_i, \mathbf{y}_i \in \mathbb{R}^m$ refer to the transcriptomes for cell $i$. Now the between-distribution distance $\delta_{\mathcal{X}\mathcal{Y}}$ and the within-distribution distances $\sigma_\mathcal{X}$ and $\sigma_\mathcal{Y}$ can be defined as:

$$\delta_{\mathcal{X}\mathcal{Y}} = \frac{1}{n_a \cdot n_b} \sum_{i=1}^{n_a} \sum_{j=1}^{n_b} d(\mathbf{x}_i, \mathbf{y}_j)$$

$$\sigma_\mathcal{X} = \frac{1}{n_a^2} \sum_{i=1}^{n_a} \sum_{j=1}^{n_a} d(\mathbf{x}_i, \mathbf{x}_j) \tag{F1}$$

$$\sigma_\mathcal{Y} = \frac{1}{n_b^2} \sum_{i=1}^{n_b} \sum_{j=1}^{n_b} d(\mathbf{y}_i, \mathbf{y}_j)$$

where $d(\cdot, \cdot)$ is the squared Euclidean distance. The E-distance, $E$, is then defined as:

$$E(\mathcal{X}, \mathcal{Y}) := 2\delta_{\mathcal{X}\mathcal{Y}} - \sigma_\mathcal{X} - \sigma_\mathcal{Y} \tag{F2}$$

The E-test, a Monte Carlo permutation test, is used to assess the statistical significance of observed E-distances. This test generates a null distribution by randomly permuting perturbation labels 10,000 times, comparing the observed E-distance against this distribution to yield an adjusted p-value that was calculated using the Holm-Sidak method. This p-value can then be used to select which perturbations result in a perturbation effect that is significantly different from the control.

Before E-statistics are calculated, the data is pre-processed. The number of cells per perturbation is subsampled to 300, following the 200-500 range proposed by Peidli et al. (2024). Perturbations with fewer than 300 cells are excluded from downstream analysis. This threshold excludes 20 perturbations, leaving 84 single-gene perturbations. One additional perturbation (*BCL2L11*) is excluded by the E-test as not significant.

For significant perturbations, the top 20 differentially expressed genes between perturbation and control are selected for evaluation. This approach is based on the observation that genetic perturbations tend to significantly affect only a fraction of the full transcriptome, while the remainder remains close to control expression (Nadig et al., 2024). This allows us to evaluate whether the predicted perturbation effect aligns with the experimental observations specifically for individual perturbations. The data is pre-processed for differential gene expression testing as described in Appendix A.2. Differential gene expression calculation is performed using the Wilcoxon rank sum test implemented in `scanpy.tl.rank_gene_groups`.

# G    CONTEXTUAL ALIGNMENT

## G.1    CALCULATING CONTEXTUAL ALIGNMENT BETWEEN PRE-TRAIN AND FINE-TUNE DATASETS

To evaluate the influence of pre-training on the efficacy of scFM embeddings, we estimate the contextual alignment between the datasets used for pre-training and those used for fine-tuning. We expect that enhanced model performance correlates with a greater overlap between these datasets. Following the instructions outlined on the scGPT GitHub, we obtained the complete pre-training cell corpus for scGPT from the CellXGene Census. As for scBERT, the pre-training dataset is derived from PanglaoDB and provided by the authors. The scBERT and scGPT datasets contain 1.4 million and 33 million cells, and 16,906 and 60,664 features respectively.

To carry out the contextual alignment experiment, we first ensure alignment between the paired datasets based on common genes. We normalize the fine-tuning dataset to a total read count of 10,000 over all genes and apply log1p-transformation. Additionally, we filter the data to include the same set of 2,061 highly variable genes that are used in the fine-tuning process (see Appendix A.2). Following these steps, we obtain two pre-training/fine-tuning common gene sets, 1,408 for scBERT + Norman and 2,044 for scGPT + Norman.

To quantify the alignment, we compare gene expression profiles between the fine-tuning and pre-training datasets by computing cosine similarity scores, which are advantageous due to their insensitivity to expression magnitude. This comparison generates a dense score matrix of dimensions $N_{\text{finetune}} \times N_{\text{pre-train}}$. For a subset of $N_{\text{pre-train}}$, used in at least one train-test split, an aggregate cross-split overlap is calculated to evaluate the impact of different pre-training/fine-tuning dataset configurations on model performance.

Initially, a matrix $S \in \mathbb{R}^{N_{\text{finetune}} \times N_{\text{pre-train}}}$ is constructed, where each element $s_{ij}$ represents the cosine similarity between the $i$-th cell in the fine-tuning dataset and the $j$-th cell in the pre-training dataset. From this, we derive a binary similarity matrix $B$ of the same dimensions with entries $b_{ij}$. The matrix is constructed as follows:

$$b_{ij} = \begin{cases} 1 & \text{if } s_{ij} \geq \mu + 2\sigma, \\ 0 & \text{otherwise,} \end{cases} \tag{G1}$$

where $\mu$ and $\sigma$ are the mean and standard deviation of the cosine similarities computed across 100,000 randomly sampled cell pairs. Based on this established threshold, $B$ represents whether each fine-tuning cell significantly overlaps with each pre-training cell.

To quantify the alignment for each fine-tuning cell, we aggregate over the pre-training dimension of matrix $B$ for each fine-tuning cell, resulting in a vector $\mathbf{f}$ where each component $f_i$ is given by:

$$f_i = \frac{1}{N_{\text{pre-train}}} \sum_{j=1}^{N_{\text{pre-train}}} B_{ij} \tag{G2}$$

Here, $f_i \in \mathbb{R}^{N_{\text{finetune}}}$ represents the fraction of the pre-training dataset that is similar to the $i$-th fine-tuning cell.

To conduct the sensitivity analysis, we define a threshold $\tau$, which represents the minimum fraction of the pre-training dataset that a fine-tuning cell must be similar to in order to be considered significantly aligned. $\tau$ is varied within the range of 0 to 0.1% of $N_{\text{pre-train}}$. For each value of $\tau$, we calculate the proportion of fine-tuning cells that meet or exceed this threshold, thus generating a series of values:

$$p(\tau) = \frac{1}{N_{\text{finetune}}} \sum_{i=1}^{N_{\text{finetune}}} \mathbf{1}_{\{f_i > \tau\}} \tag{G3}$$

where $\mathbf{1}$ is the indicator function that evaluates to 1 if the condition is true and 0 otherwise.

The sensitivity curve is then plotted as $p(\tau)$ versus $\tau$. The area under this curve reflects the overall cross-split overlap of the fine-tuning dataset relative to the pre-training dataset, as visualized in Figure G1.

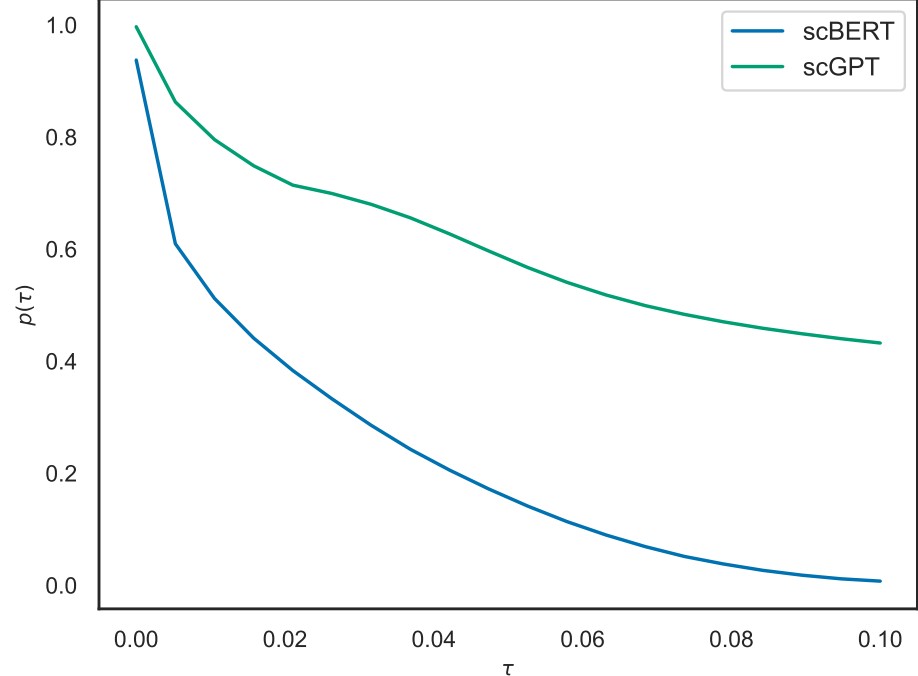

Figure G1: Plot of the probability that a cell from the pre-train dataset is similar to a cell from the fine-tune dataset as a function of $\tau$, the similarity threshold at which two cells are considered similar based on their cosine similarity.

# H SUPPLEMENTARY FIGURES

## H.1 SPECTRA PERFORMANCE CURVES

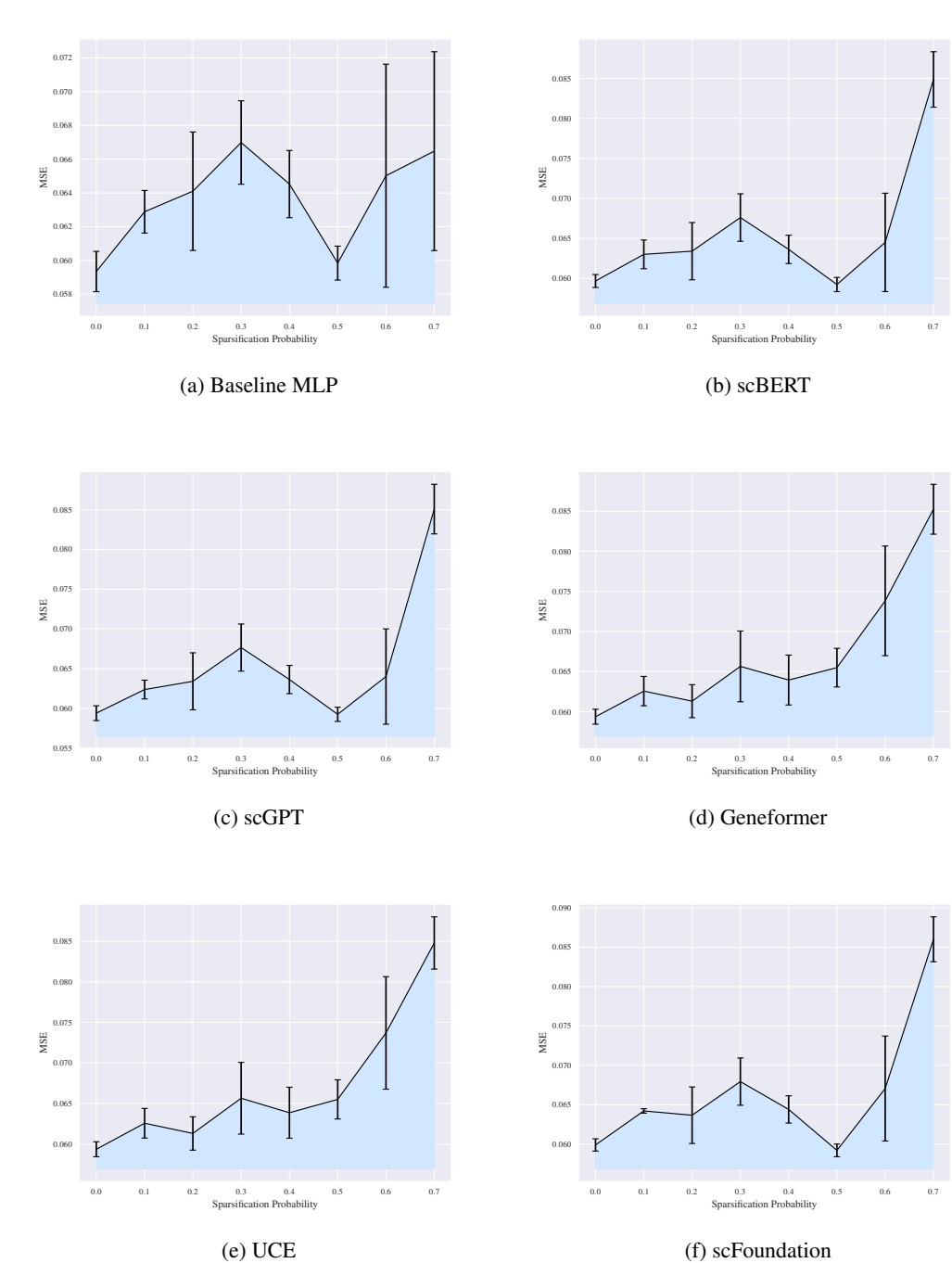

Figure H1: MSE as a function of the sparsification probability for the different models. These functions are used to calculate to calculate the AUSPC, which is here shaded in blue.

## H.2 Perturbation effect prediction results across top 20 DEGs

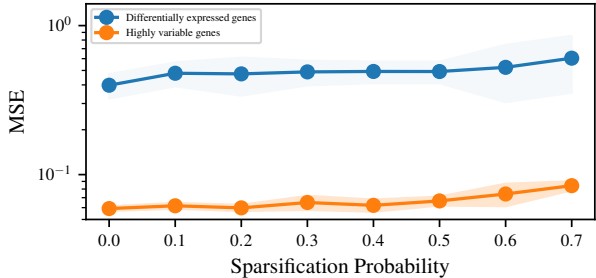

Figure H2: Comparison of the mean baseline across different sparsification probability train-test splits.

## H.3 MSE for all models compared to mean baseline across top 20 DEGs

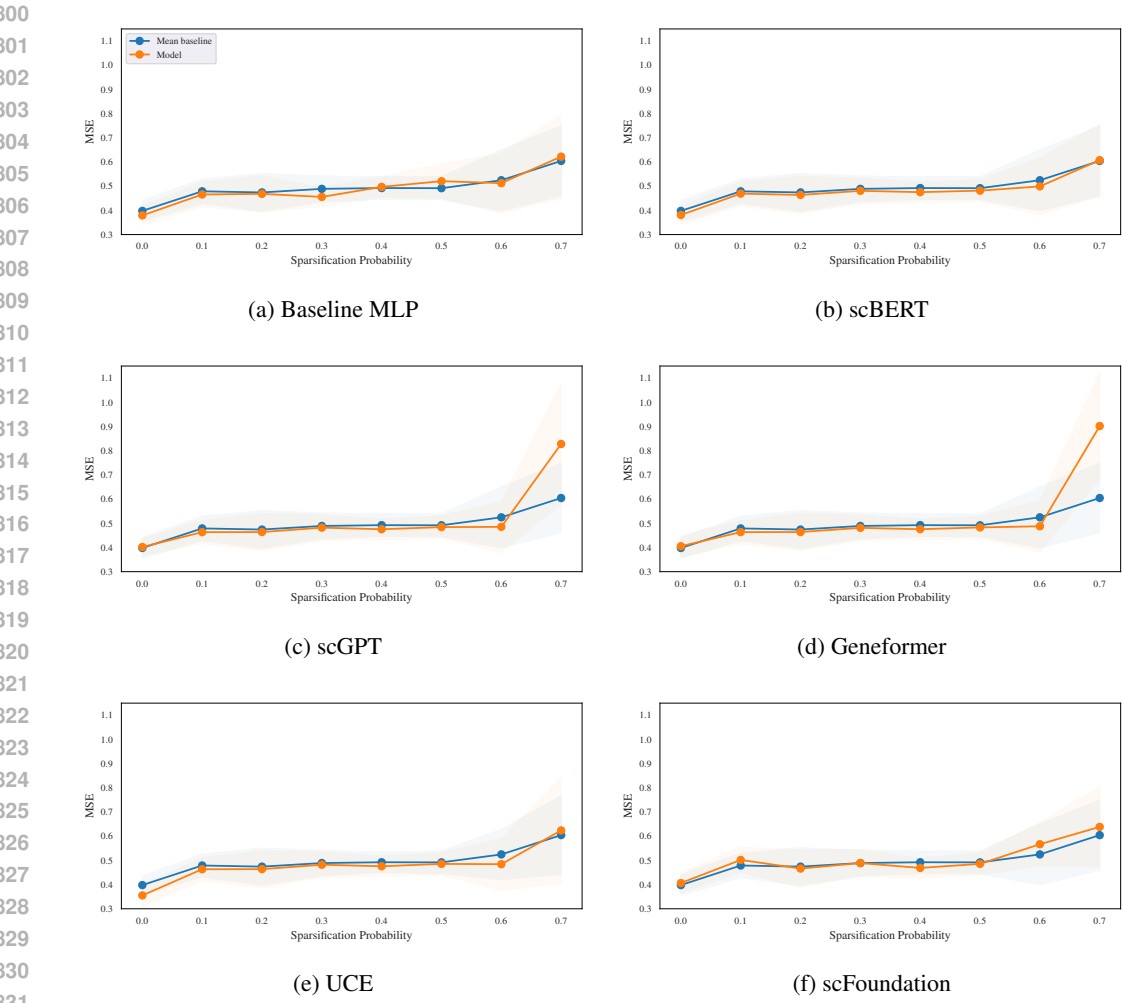

(a) Baseline MLP

(b) scBERT

(c) scGPT

(d) Geneformer

(e) UCE

(f) scFoundation

Figure H3: MSE as a function of the sparsification probability for the different models. This is a depiction of the curves that are used to calculate the $\Delta$AUSPC.

## H.4 MEAN POST-PERTURBATION EXPRESSION PROFILES FOR *IKZF3* AND *CEBPA*

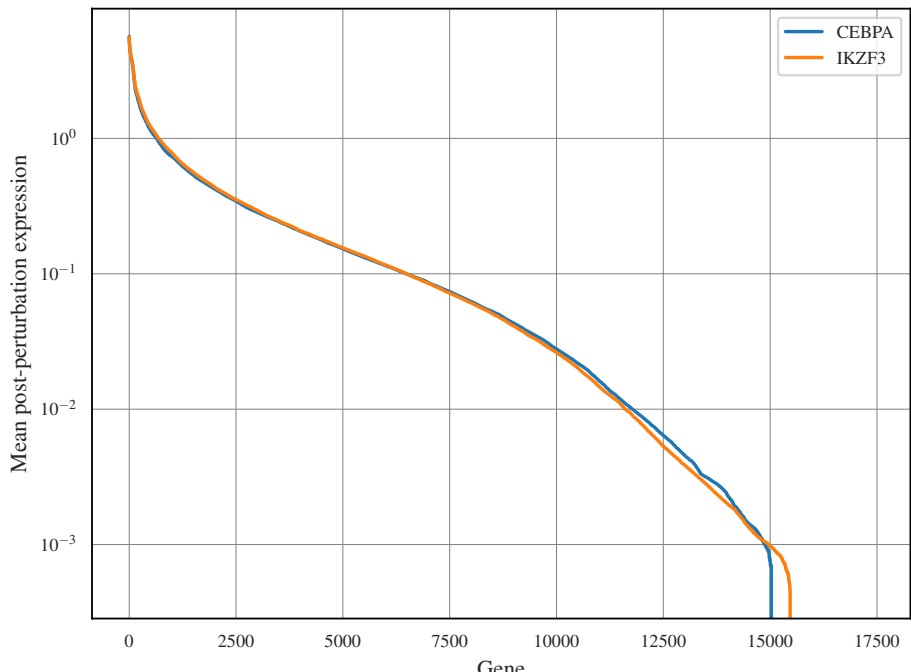

Figure H4: Post-perturbation mean expression profiles for *IKZF3* and *CEBPA*. The y-axis has been log-transformed for visual clarity.

