# OpenReview forum: "PertEval-scFM: Benchmarking Single-Cell Foundation Models for Perturbation Effect Prediction"
_ICLR.cc/2025/Conference — Submitted to ICLR 2025_

### Official Review · Reviewer_yfBT · 2024-11-02

**Soundness:** 2
**Presentation:** 3
**Contribution:** 1
**Rating:** 3
**Confidence:** 4

**Summary:**

The paper introduces an evaluation framework for scRNA-seq foundation models applied to the downstream task of predicting the transcriptomic effect of unseen genetic perturbations. In order to adapt foundation models to this task, the authors first extract gene embeddings from a variety of foundation models by simulating a genetic knock-out in unperturbed cells and then embedding both unperturbed and in-silico perturbed cells using the foundation model. Then, they fit MLP adapters on top of these embeddings with a root mean squared error loss. They include two baselines to compare against: the mean baseline, (presenting no variation between different perturbations) and an MLP adapter fit on a simple co-expression based embedding.

The authors then proceed to evaluate these models. At its heart, the proposed evaluation suite relies on two previously established metrics, the mean squared error (MSE) over the top 20 differentially expressed genes (DEGs) per perturbation and the MSE over 2000 highly variable genes (HVGs). On top of this, a spectral splitting technique (SPECTRA) is applied to simulate the effects of increasingly stronger distributional shifts between train and test data, and the top 2000 HVG MSE is averaged over different splitting strengths to yield the area under the SPECTRA performance curve (AUSPC) metric. The authors overall find that foundation model embeddings show a marginal, but not stastitically significant improvement over simple baselines in these metrics.

Finally, they investigate properties of individual perturbations and of the different foundation models used that could explain the observed performance. First, they show that larger effect sizes as measured by the energy distance of the perturbation to the control population and uncommon effect distributions are correlated with higher MSE per perturbation. Second, they explore the effect of overlap between foundation model pre-training corpus and the downstream perturbation dataset on the overall performance, finding mostly a correlation with robustness at a very stringent data splitting regime, with little overall impact otherwise.

**Strengths:**

* **Impactful question:** Predicting the effect of unseen perturbations is at the heart of understanding the causal impact of genes and therefore an important biological question. Therefore, the scope of the paper, evaluating the impact of scRNA-seq foundation models on this task, is an important research topic.
* **Manuscript quality:** The manuscript is overall well-written and clearly organized.
* **Novelty:** The two elements that are highlighted as novel, simulating challenging datasets with SPECTRA and investigating contextual alignment, are, to my knowledge, novel in this domain.
* **Publishing negative findings:** The authors overall report negative findings as to the impact of foundation models on the task in question, which is is often  disincentivized by the scientific community. I find this commendable since it can help the community to make progress and improve these tools.

**Weaknesses:**

* **Benchmarking on inappropriate dataset**
    * Most importantly, the authors rely on a **single and quite small** (<300 overall perturbations) perturbation dataset for all of their evaluation, namely [Norman (2019)](https://www.science.org/doi/full/10.1126/science.aax4438). Despite the overall solid presentation, this leaves the impression of a rushed and unfinished work. At the very least, I would like to see results on [Replogle (2022)](https://www.cell.com/cell/fulltext/S0092-8674(22)00597-9?dgcid=raven_jbs_aip_email) included (>10,000 perturbations), but a comprehensive evaluation framework should ideally look at most available Perturb-seq datasets out there to be able to draw any insightful conclusions.
    * More than half of the perturbations in Norman (2019) are combinations of genetic perturbations. Figure E1 shows that the simplest data splits contain >200 perturbations, suggesting that these combinations were included in the analysis. However, I don't see any mention of how combinations are handled by the MLP adapters, and the authors never show any evaluation pertaining to the effect of combinations, as done, for example, in [Roohani (2024)](https://www.nature.com/articles/s41587-023-01905-6).
    * The Norman et al dataset contains CRISPRa perturbations, resulting in overexpression of the perturbed genes, i.e., a **gain-of-function perturbation**. The foundation model embeddings employed by the authors simulate **loss-of-function perturbations**. On a biological level, this is highly unintuitive. While it may still lead to meaningful and useful embeddings, this is a potential failure mode that could explain the overall findings of the paper.
* **Dataset splitting is not high-impact:** Since evaluation is performed on individual datasets (or more specifically, a single dataset here), generalizing to unseen perturbations could be construed as an experimental design question: can I generalize from the observed perturbations to unseen ones, and if so, which ones should I collect data for? Since Perturb-seq can already applied to the order of thousands of perturbations, if the task is to generalize to arbitrary perturbations, a **random selection of genes is always an admissible design**. While novel, that means that making the prediction task artifically harder through the SPECTRA split might overestimate the difficulty of the problem in an actual experimental design context. Moreover, the dataset that the authors consider is already very small and is further massively reduced (see Figure E1, less than 30 perturbations in the final split). Overall, I am not convinced that this splitting technique should be an integral part of the eval pipeline.
* **Correlative investigations of model performance are not convincing**
    * The observed correlation of E-distance to MSE (Figure 3a) is interesting, but not completely convincing to me. At a minimum, could the authors compute a measure of correlation in this plot? Moreover, it should be able to not only assess this question observationally, but also causally through ablation studies. Does changing the splits to specifically include similar E-distance perturbations change the observed performance? Overall, all this seems to say is that outliers are harder to predict.
    * Similarly, the different contextual alignment examples in Figure 5, while novel, do not actually show any good stratification of results (except for the 0.7 split which is very small, and even there it might be exacerbated by the box-plot excluding outliers for scBERT). Therefore, I'm not convinced as to why this is the right formulation to consider for assessing how appropriate the pre-training dataset is for the downstream task.
* **Summary and rating:** The above weaknesses by far outweigh the strenghths, leading to a recommendation to **reject**.

**Questions:**

* It is not clear to me how the mean baseline is computed. Is it the mean over all perturbations? This is not immediate from the notation in Equation (6).
* The error bars in Figure 2 are very large. Could this be improved by averaging over more model splits?
* In Figure 3, how are the MSEs per perturbation calculated over splits? Are they averaged here? Would it make more sense to investigate this on a per split basis?

---

> ### Author Response · Authors · 2024-11-21
> **Dataset**
>
> We thank the reviewer for their detailed feedback and for recognising the strengths of our work, including the importance of the research question, the overall quality of the manuscript, the novelty of SPECTRA and contextual alignment, and our commitment to publishing negative findings. Below, we address the concerns and questions raised in detail.
>
> **Benchmarking on a single dataset**
> We agree that the use of more datasets would broaden the scope of our analysis. We want to clarify that we originally focused on the Norman dataset because it has been shown to contain the strongest perturbation effect signal for genetic perturbations as measured by the E-distance [See Figure 3.b. in [1]] and is therefore best suited to our analysis. We hypothesised that if scFM embeddings contained biologically relevant information for perturbation effect prediction, it would be more likely to be contained in a dataset with a strong perturbation effect signal. Positive results in this context would indicate that biologically relevant perturbation effects can be predicted accurately with zero-shot scFM embeddings, while negative results would provide greater confidence that the observed limitations reflect true negatives, rather than being due to low-quality signal inherent to the dataset tested. In line with this argument, the Replogle dataset, despite containing a greater number of perturbations (~4,000 K562; ~200 RPE1), demonstrates much diminished perturbation effect signal overall [See Figure 3.b. in [1]]. We have expanded our analysis to include all the combinatorial perturbations in the Norman dataset, where originally we only considered single-gene perturbations. We present the updated results in Table 3 for the top 2000 HVGs and in Table 4 for the top 20 DEGs.
>
> **Table 3: Perturbation effect prediction evaluation across 2000 HVGs.**
> |Model|Perturbation strategy|SP0.0|SP0.1|SP0.2|SP0.3|SP0.4|SP0.5|SP0.6|SP0.7|AUSPC|ΔAUSPC|
> |-----|--------------------|-----|-----|-----|-----|-----|-----|-----|-----|-----|------|
> |Mean baseline|Dual-gene|5.337±0.093|5.257±0.102|5.910±0.255|5.722±0.401|6.644±0.167|7.674±0.962|6.071±0.772|5.201±0.594|4.255±0.073|-|
> |MLP gene expression|Dual-gene|5.337±0.094|5.261±0.100|5.913±0.255|5.728±0.402|6.635±0.161|7.675±0.953|6.050±0.763|5.198±0.593|4.253±0.073|0.002|
> |scFoundation|Dual-gene|5.675±0.106|5.564±0.051|6.173±0.196|6.050±0.462|6.755±0.279|7.944±1.186|6.382±0.876|5.238±0.578|4.432±0.467|-0.177|
> |UCE|Dual-gene|5.655±0.091|5.514±0.066|6.145±0.183|6.029±0.460|6.736±0.289|7.939±1.184|6.352±0.831|5.612±0.665|4.435±0.085|-0.180|
> |scBERT|Dual-gene|5.655±0.092|5.515±0.067|6.159±0.196|6.022±0.465|6.757±0.281|7.999±1.240|6.493±0.736|7.110±0.579|4.533±0.081|-0.278|
> |scGPT|Dual-gene|5.654±0.091|5.515±0.067|6.153±0.189|6.023±0.464|6.766±0.287|8.272±1.377|8.826±0.182|14.906±2.154|5.184±0.132|-0.929|
>
> **Table 4: Perturbation effect prediction evaluation across the top 20 DEGs.**
>
> *Note: For two-gene perturbations split 0.5, there were not enough perturbations that passed our quality control to properly define the split.*
> |Model|Perturbation strategy|SP0.0|SP0.1|SP0.2|SP0.3|SP0.4|SP0.5|SP0.6|SP0.7|↓AUSPC(10⁻²)|↑ΔAUSPC(10⁻²)|
> |-----|--------------------|-----|-----|-----|-----|-----|-----|-----|-----|-----------|-------------|
> |Mean baseline|Dual-gene|0.079±0.033|0.091±0.041|0.078±0.038|0.059±0.032|0.091±0.042|-|0.038±0.014|0.095±0.044|0.051±0.005|-|
> |MLP gene expression|Dual-gene|0.053±0.001|0.053±0.001|0.059±0.002|0.058±0.003|0.068±0.001|-|0.062±0.008|0.054±0.005|0.042±0.001|0.900|
> |scFoundation|Dual-gene|0.493±0.044|0.534±0.057|0.554±0.070|0.606±0.073|0.656±0.045|-|0.621±0.060|0.497±0.051|0.410±0.008|-35.9|
> |scGPT|Dual-gene|0.489±0.043|0.527±0.056|0.550±0.069|0.597±0.072|0.673±0.044|-|0.724±0.028|1.941±0.329|0.499±0.018|-44.8|
> |UCE|Dual-gene|0.489±0.043|0.527±0.055|0.550±0.069|0.601±0.072|0.656±0.043|-|0.624±0.053|0.506±0.048|0.410±0.007|-35.9|
> |Geneformer|Dual-gene|0.489±0.043|0.527±0.055|0.550±0.069|0.603±0.076|0.661±0.045|-|0.623±0.054|0.487±0.048|0.409±0.008|35.8|
> |scBERT|Dual-gene|0.490±0.043|0.528±0.056|0.550±0.069|0.596±0.071|0.661±0.041|-|0.622±0.049|0.681±0.086|0.418±0.008|-36.7|
>
> [1] Peidli et al., 2024. scPerturb: harmonized single-cell perturbation data. https://www.nature.com/articles/s41592-023-02144-y

---

> ### Author Response · Authors · 2024-11-21
> **Dataset (continued)**
>
> From the results presented in Tables 3 and 4, we observe a slight increase in prediction performance across certain models. However, the relative performance compared to baseline models diminishes as task complexity increases. This aligns with our prior observations that generalisability decreases for perturbations that are strong or atypical, which tend to be overrepresented in dual-gene perturbations. While the additional results provide further evidence of the trends observed in our initial analysis, they also underline the inherent limitations of current scFM embeddings in addressing the complexities of dual-gene perturbations and emphasise the necessity of future efforts to improve their robustness and biological relevance. Furthermore, we are currently working on including the Replogle et al. dataset in our benchmark. While compute and memory constraints currently pose challenges to processing this dataset, we are working on engineering solutions to implement this analysis and aim to report these results as soon as they are available.

---

> ### Author Response · Authors · 2024-11-21
> **Biological relevance of loss-of-function simulations**
>
> **Biological relevance of loss-of-function simulations**
>
> We thank the reviewer for highlighting the potential limitations of simulating perturbations by nullifying the expression of the perturbed gene to generate scFM embeddings. While we acknowledge that this approach may appear counterintuitive from a biological perspective, it was chosen to address the challenges inherent in standardising perturbation simulations across foundation models. scFMs differ in gene expression representation. For instance, scGPT encodes gene expression as numerical values, whereas Geneformer uses rank-order representations.  If we were to simulate upregulation by artificially increasing a gene's expression value in numerically encoded models, there is no biologically grounded way to determine the appropriate magnitude. Arbitrary modifications risk introducing biases that are inconsistent across models. For non-parametric models that use rank-order representations, simulating upregulation becomes even more ambiguous, as it is unclear how to re-rank genes to reflect such a perturbation.
>
> On the other hand, nullifying the expression of a perturbed gene provides a uniform and unbiased approach that avoids model-specific inconsistencies, enabling fairer comparisons across diverse foundation models. While this approach is better suited to loss-of-function scenarios, we recognise its limitations in representing gain-of-function perturbations, such as those present in the Norman dataset. This limitation reflects a broader challenge in foundation model methodologies, which we briefly address in the discussion section of the paper: “*key questions, such as how to represent perturbations *in silico*, \[...\] need to be addressed*” (line 469). Addressing these representational issues will be crucial for future development of foundation models to better capture the effects of both loss- and gain-of-function perturbations. Despite this, we believe that it is beyond the scope of our benchmarking paper to suggest alternative approaches for this issue. We hope our work will motivate further efforts to refine how perturbations are simulated to improve their biological relevance and predictive accuracy.

---

> ### Author Response · Authors · 2024-11-21
> **Dataset splitting and SPECTRA**
>
> **Dataset splitting and SPECTRA**
>
> We appreciate the concern regarding the impact of dataset splitting. However, we believe SPECTRA’s contribution extends beyond what random splits or current experimental designs offer. In real-world applications like Perturb-seq, unpredictable biological distribution shifts are common. Random splits are not inherently easier or harder; a random split could align with a sparsification	probability of 0.1 or 0.7, representing significantly different train-test dissimilarities. However, systematically quantifying this dissimilarity makes it possible to contextualise performance results and compare model robustness across different tasks or datasets. Random splits may therefore fail to capture these distribution shifts, often leading to overly optimistic assessments of model performance. SPECTRA addresses this by generating splits of increasing difficulty, starting with random splits and progressing to more challenging ones based on train-test similarity. This approach ensures that SPECTRA encompasses random splits but also evaluates a model’s robustness across a broader spectrum of biological variability. SPECTRA does not artificially inflate task difficulty but rather controls for the similarity between train and test split. This indeed increases test difficulty, but not artificially. While controlling for train-test similarity may lead to smaller datasets in harder splits, this is a reflection of the redundancy inherent to the dataset, not a limitation of SPECTRA. Moreover, SPECTRA is not just an evaluation tool but a valuable guide for experimental design. For example, in Perturb-seq, SPECTRA can reveal if a model generalises well within pathways but struggles across pathways, guiding researchers on which genes to test next. Random splits, by contrast, obscure such insights due to high train-test similarity. We argue that SPECTRA’s ability to highlight weaknesses in generalisability, simulate real-world distribution shifts, and inform experimental design makes it a high-impact addition to evaluation pipelines, especially for tasks like Perturb-seq where navigating distribution shifts is critical.

---

> ### Author Response · Authors · 2024-11-21
> **Correlative investigations of model performance**
>
> **Correlative investigations of model performance**
>
> We thank the reviewer for pointing our potential improvements to Figure 3. To address these concerns, we have updated Figure 3 (see updated manuscript, Figure 3a and 3b), where we have included the Spearman correlation coefficient to quantify these observed relationships and we have added two-gene perturbations to increase the statistical power of the analysis. The results indicate that prediction performance does degrade as E-distance increases, but the relationship is weak ($\text{Spearman’s R} = 0.13, \text{p} = 1.22 × 10^{⁻⁸}$).  This suggests that the model’s capability to predict perturbation effects depends not only on the magnitude of the perturbation, but also on its distribution. For example, we observed that some pronounced perturbations, such as CEBPA, are well-predicted despite having a high E-distance. This shows how we can use the E-distance to pinpoint interesting perturbations and failure modes beyond perturbation strength.
>
> The reviewer notes that the figure “*seems to say outliers are harder to predict*”, but this interpretation is not entirely accurate. As shown in Figure 3a, the model performed well on certain perturbations, such as CEBPA, despite being one of the most pronounced perturbations in the dataset, as measured by E-distance. Figure 4 provides insight into this apparent discrepancy, suggesting that the narrow peak of the effect inflates the E-distance while the long tail of minimal effects makes the overall distribution easier to predict.

---

> ### Author Response · Authors · 2024-11-21
> **Clarifications and conclusion**
>
> **Clarifications**
>
> **Mean baseline (eq. 6):** The mean baseline is computed as the mean expression profile across all unperturbed cells for each gene. We will revise the manuscript to clarify this computation more explicitly.
>
> **Error bars in Figure 2:** The large error bars reflect variability across splits and model replicates. Increasing the number of splits and replicates could reduce this variability, and we note that the error bars seem large in the context of the of the scFMs results as they perform similarly, comparing the errors in GEARS with the scFMs for the first spectral splits we see that they do not overlap. Thus, the overlapping errors are a reflection of the similar performance of the scFMs rather than the number of splits.
>
> **MSE per perturbation (Figure 3):** The results reported in Figure 3 are on a per-split basis. Specifically, we averaged the results across foundation models per split. This is apparent from the legend in which we indicate through colour coding what data point belongs to what split.
>
> **Conclusion**
>
> We thank the reviewer for their thoughtful feedback, which has highlighted several areas for improvement. We have increased the power of our dataset by including Norman two-gene perturbations and are addressing dataset limitations by incorporating the Replogle dataset and plan to expand our benchmark further over time. Thank you again for your constructive comments.

---

> ### Comment · Reviewer_yfBT · 2024-11-21
> **Re: Clarifications and conclusion**
>
> > MSE per perturbation (Figure 3): The results reported in Figure 3 are on a per-split basis. Specifically, we averaged the results across foundation models per split. This is apparent from the legend in which we indicate through colour coding what data point belongs to what split.
>
> I'm still not sure I understand this: is there only a single train-test split per sparsification probability? And every perturbed gene only appears in a single split and sparsification probability? Else, how did you pick which sparsification probability to assign to a gene in Figure 3?

---

> ### Comment · Reviewer_yfBT · 2024-11-21
> **Re: Dataset**
>
> > dual gene results
>
> Thanks for adding this! Could you provide details on how you implemented the prediction, ideally in an updated version of the manuscript? Thanks!

---

> ### Author Response · Authors · 2024-11-22
>
> >is there only a single train-test split per sparsification probability?
>
> There are three train-test splits per sparsification probability ($s$) because we run the experiments in triplicate with different seeds. To determine these splits, we construct a similarity graph where similar perturbations are connected.  $s$ then controls how many edges are stochastically removed from the similarity graph to obtain subgraphs from which train and test perturbations can be sampled. This enforces a certain degree of cross-split overlap between train and test sets such that as $s$ increases, the train and test sets become more dissimilar. We explicitly quantify this cross-split overlap and illustrate its decline as $s$ increases (Appendix Figure E1.b). Figure 3 shows the splits across all replicates. Further details on SPECTRA, our train-test splitting procedure, and the role of the sparsification probability are provided in Section 2.2.3 and Appendix E of the manuscript.
>
> > And every perturbed gene only appears in a single split and sparsification probability?
>
> A single perturbed gene can theoretically appear across multiple splits. However, SPECTRA ensures that the same or sufficiently similar perturbations do not simultaneously appear in both the train and test sets of any split. This separation minimises information leakage between the train and test sets and allows us to simulate distribution shifts.
>
> > how did you pick which sparsification probability to assign to a gene in Figure 3?
>
> SPECTRA tracks the assignment of each perturbation to its respective train-test split. We identify a split by its sparsification probability, which runs from 0.0 to 0.7. For Figure 3 in particular, we are visualising the test perturbations for all splits and from SPECTRA we can link exactly which perturbation belongs to which train-test split. This can be appreciated in Figure 3a through the colour coding, where each test perturbation is assigned to a particular split, and in Figure 3b, we have organized all the test perturbations per split.
>
> Thank you for bringing up these points. We hope these clarifications address your questions.

---

> ### Author Response · Authors · 2024-11-22
>
> To implement the double-gene perturbations, we calculated the co-expression matrix similar to how we did for the single-gene perturbation, and averaged to obtain a unified co-expression matrix.
>
> We have now updated the manuscript to reflect this, along with the following changes:
> - Added dual-gene perturbations into Table 2, for evaluation across 2,000 Highly Variable Genes (HVGs).
> - Added dual-gene perturbations into Table 3, for evaluation across 20 Differentially Expressed Genes (DEGs).
> - Updated Figure 3 to include Spearman correlation and added all dual-gene perturbations to increase the power and statistical significance of the observed trend.
> - Included explanation on dual-gene _in silico_ perturbation representation in Section 2.1.2 and Appendix C1.1 and C2.
> - Included interpretation of the dual-gene results in Sections 3.1, 3.2 and 3.3.
> - Mentioned double-gene perturbation evaluation in the conclusion.
>
> Changes have been marked in red, so they should be easy to spot.

---

> > ### Comment · Reviewer_yfBT · 2024-11-26
> > **Thank you for your responses**
> >
> > Thank you for prompt responses and the work to onboard the dual-gene evaluation setup.
> >
> > While I appreciate the comprehensiveness on the number of evaluated foundation models, I believe that the paper is still limited by the choice of a single dataset. Given that the task considered is already pretty hard, I also consider the impact of the SPECTRA splitting strategy to be limited.
> >
> > Therefore, I maintain my current evaluation rating.

---

> ### Author Response · Authors · 2024-11-29
>
> We sincerely thank the reviewer for acknowledging our extensive efforts to include two-gene perturbations across all models and baselines, including the newly added GEARS, evaluated under both HVG and DEG setups.
>
> We appreciate the reviewer’s continued concern regarding the use of a single dataset. To address this, we are actively working on integrating the Replogle et al. (2022) dataset \[1\]. Our preprocessing and featurisation pipeline is fully operational, and model training is currently underway. We anticipate completing the full evaluation within approximately one week. As an update, we provide preliminary results for three scFMs and two baselines across five SPECTRA splits, which highlight the progress we have made in broadening our dataset scope.
>
> Regarding the SPECTRA splitting strategy, we respectfully maintain that its value extends beyond merely increasing task difficulty. SPECTRA provides a systematic and fair approach to evaluating the generalisability of methods under varying levels of distribution shift, a critical aspect in the context of single-cell datasets, which are particularly prone to batch effects and noisy measurements. The importance of assessing robustness to such shifts is further highlighted by the performance of task-specific models like GEARS. While GEARS outperforms other methods in general, it still exhibits a clear degradation in performance as sparsification probability increases. Furthermore, differences in the performance of embeddings from individual scFM models further underscore the relevance of this analysis. For example, scGPT experiences catastrophic performance degradation at a sparsification probability of 0.7, whereas UCE maintains performance levels comparable to the baseline (Appendix H.3). These insights are valuable for understanding the robustness and limitations of the evaluated models and are independent of the inherent complexity of the task itself. We hope this perspective clarifies the importance of SPECTRA in the broader context of model evaluation.
>
> Thank you once again for your thoughtful feedback.
>
> \[1\] Replogle, J. M. et al. Mapping information-rich genotype–phenotype landscapes with genome-scale Perturb-seq. Cell 185, 2559–2575 (2022).

---

### Official Review · Reviewer_3E1k · 2024-11-03

**Soundness:** 3
**Presentation:** 4
**Contribution:** 4
**Rating:** 6
**Confidence:** 3

**Summary:**

This paper introduces PertEval-scFM, a framework for evaluating how well single-cell foundation models (scFMs) can predict gene expression changes in response to genetic perturbations. The authors benchmark zero-shot embeddings from various scFMs against simple baseline models, finding that scFMs don't consistently outperform baselines. They indentify narrow training distributions as one of the potential culprits and suggest improvements.

**Strengths:**

- Creation of a standardized evaluation framework (PertEval-scFM) with well-documented codebase is good for field progress
- I like that there's multiple complementary evaluation methods (E-distance, AUSPC), including some novel metrics
- Evaluation includes all major single-cell foundation models with appropriate baselines
- I like how the paper provides constructive criticism of current models while suggesting concrete paths forward, by doing the distribution shift effect analysis.

**Weaknesses:**

- Limited dataset scope: Heavy reliance on single dataset (Norman et al.), restricted to CRISPRa perturbations in K562 cell line make me concerned about how well this generalizes
- My main issue with the paper is that there seem to be big parameter count disparaties betw. the MLPs because of the different embedding dimensions.
        - Base embedding dimensions vary widely (200 to 3,072)
        - Results in vastly different MLP sizes between models - by my calculations, scFoundation's MLP ends up with around 25M parameters while scBERT's has less than 0.5M
        - Such dramatic differences in model capacity could significantly affect training dynamics and make fair comparison difficult and this doesn't seem significantly considered / addressed. Though the authors tuned the hypers individually for each embedding.
- Minor note: In Table 1 scGPT is a Transformer, FlashAttention is just a way of efficiently implement normal attention.

**Questions:**

1. Could you elaborate on the methodological choices around MLP probe architecture? Specifically:
    - How sensitive are results to the choice of architecture?
    - How does the varying parameter count due to different embedding dimensions affect fair comparison between models?
    - Have you considered approaches to normalize model capacity across different embeddings?

---

> ### Author Response · Authors · 2024-11-21
> **Limited dataset scope**
>
> We thank the reviewer for their thoughtful feedback and for recognizing the strengths of our work, including the creation of a standardised evaluation framework, the inclusion of complementary and novel metrics, the evaluation of major scFMs with appropriate baselines, and the constructive paths forward suggested by our distribution shift analysis. Below, we address the reviewer’s concerns in detail:
>
> **Limited dataset scope**
>
> Regarding the use of additional datasets, we agree this would broaden the scope of our analysis. We want to clarify that we originally focused on the Norman dataset because it has been shown to contain a higher-quality and stronger perturbation signal \[See Figure 3.b. in [1](https://www.nature.com/articles/s41592-023-02144-y/figures/3)\] and was therefore better suited to our analysis. We hypothesised that if scFM embeddings contained biologically relevant information for perturbation effect prediction, it would be more likely to manifest in a dataset with a strong signal like Norman. Positive results in this context would indicate that biologically relevant perturbation effects can be predicted accurately with zero-shot scFM embeddings, while negative results would provide greater confidence that the observed limitations reflect true negatives, rather than being due to low-quality signal inherent to the dataset tested.
>
> In line with this argument, the Replogle dataset, despite containing a greater number of perturbations (\~4,000 K562; \~200 RPE1), demonstrates a lower-quality perturbation signal overall \[[1](https://www.nature.com/articles/s41592-023-02144-y)\]. This may be due in part to the complexity of its perturbation space: seeing more perturbations simultaneously may introduce signal that is diffuse and not conclusively tied to any single perturbation, resulting in increased noise. This characteristic makes it less well-suited for drawing strong conclusions about scFM embeddings in their current form. However, we agree with the reviewers that incorporating additional datasets like Replogle would strengthen the paper and increase the credibility of our conclusions. While compute and memory constraints currently pose challenges to processing the Replogle dataset, we are working on engineering solutions to enable this analysis. We are committed to reporting these results as soon as they are available.
>
> [1] Peidli et al., 2024. scPerturb: harmonized single-cell perturbation data. https://www.nature.com/articles/s41592-023-02144-y

---

> ### Author Response · Authors · 2024-11-21
> **Parameter count disparities in MLP probes**
>
> **Parameter count disparities in MLP probes**
>
> We appreciate the reviewer’s concern about the disparities in MLP parameter counts due to varying embedding dimensions across scFMs. We acknowledge that this could influence training dynamics and make fair comparison challenging. To alleviate this concern, we trained a range of MLPs with increasing parameter count on our raw expression data to verify the effect of parameter number on results. We report our findings in **Table 1**, where it can be seen the increase in parameters has no effect on the MSEs obtained. We also include additional results using scBERT and scFoundation embeddings as input, with increasing parameter count. This further strengthens our conclusion that, despite an increase in the expressiveness of the underlying MLP model, the scFM embeddings do not contain sufficient biologically relevant information from the pre-training stage to provide increased performance for this task.
>
> **Table 1: Train and Test set results with MLPs of increasing parameter count**
>
> | Trainable parameters (million) | Training data             | train/MSE   | val/MSE   |
> |--------------------------------|---------------------------|-------------|-----------|
> | 1.6                            | Raw gene expression       | 0.057067    | 0.057642  |
> | 3.2                            | Raw gene expression       | 0.058670    | 0.057493  |
> | 6.3                            | Raw gene expression       | 0.056748    | 0.057424  |
> | 12.7                           | Raw gene expression       | 0.056724    | 0.057428  |
> | 1.6                            | scFoundation embeddings   | 0.060780    | 0.060260  |
> | 3.2                            | scFoundation embeddings   | 0.060440    | 0.059910  |
> | 12.6                           | scFoundation embeddings   | 0.059570    | 0.059050  |
> | 0.2                            | scBERT embeddings         | 0.061040    | 0.061426  |
> | 1                              | scBERT embeddings         | 0.061046    | 0.061428  |
> | 8                              | scBERT embeddings         | 0.061040    | 0.061421  |
>
> We agree that normalising model capacity — for instance, by using fixed-capacity projection layers to map all embeddings to a standardised dimensionality before passing them into the MLP — could theoretically provide a more equitable comparison. However, the results in Table 1 suggest that such normalisation is unlikely to change the overall conclusions. Additionally, since we are primarily evaluating the intrinsic information content of the embeddings, projecting them to a different dimensionality might introduce confounding factors, potentially obscuring the original information encoded in the embeddings.
>
> Despite the disparities in parameter count, our current approach maintains alignment with the standard practice of tailoring MLP architectures to the embedding size to ensure sufficient expressiveness for each model. This guarantees that no embedding is artificially constrained or unfairly advantaged due to mismatched architecture capacity.

---

> ### Author Response · Authors · 2024-11-21
> **Other concerns and conclusion**
>
> **Methodological choices around MLP probes**
>
> We selected simple MLP probes with one hidden layer to isolate and evaluate the intrinsic information content of scFM embeddings. This minimalistic architecture reduces confounding effects from complex prediction heads and focuses on the utility of the embeddings themselves.
>
> **Sensitivity to architecture choices:** Our experiments varying the number of layers and hidden dimensions (see Table 1\) showed no significant improvement in performance. This suggests that the limitations lie in the embeddings themselves rather than the probe architecture.
>
> **Minor Note on Table 1**
>
> This has now been changed to “Transformer” in Table 1, thank you for spotting that\!
>
> **Conclusion**
>
> We appreciate the reviewer’s insightful comments, which have highlighted important areas for improvement. We are addressing the dataset scope concern by incorporating the Replogle dataset into our benchmark and expanding the GitHub resource over time. We have investigated the impact of MLP parameter disparities and found limited sensitivity to architecture choices. Thank you again for your constructive feedback.

---

> ### Comment · Reviewer_3E1k · 2024-11-27
>
> I thank the reviewers for their detailed response and for adding additional information to the paper. Since my main problem has now been sufficiently addressed I have decided to raise my score.

---

> ### Author Response · Authors · 2024-11-29
>
> We sincerely thank the reviewer for raising our score and for the constructive feedback.
>
> To address the original concerns about dataset scope, we are actively working on incorporating the Replogle dataset [1] into our analysis. This dataset introduces approximately 2,000 perturbations, significantly expanding the scope of our evaluation. Our preprocessing and featurization pipelines for this dataset are fully operational. However, due to its size, generating the necessary features and training all models across all splits requires substantial computational resources. We anticipate completing this process within one week.
>
> In the meantime, we are pleased to share preliminary results in the updated PDF for three foundation models and two baselines across five SPECTRA splits. These results demonstrate our progress and reflect our commitment to broadening the study’s dataset scope as originally suggested.
>
> Thank you again for your valuable input and insightful comments.
>
> [1] Replogle, J. M. et al. Mapping information-rich genotype–phenotype landscapes with genome-scale Perturb-seq. Cell 185, 2559–2575 (2022).

---

### Official Review · Reviewer_SABf · 2024-11-03

**Soundness:** 3
**Presentation:** 3
**Contribution:** 2
**Rating:** 3
**Confidence:** 4

**Summary:**

This work introduces a benchmark framework for the evaluation of single-cell FMs in the context of genetic perturbation prediction. Multiple scFMs (scBERT, Geneformer, scGPT, UCE, and scFoundation) are compared to simple baselines through a linear probing approach and evaluated across multiple metrics aggregated from previous works. Results highlight the limitations of scFMs in this context, with negligible improvements over the baselines. Additionally, the authors highlight the need for better datasets across a broader and more diverse range of cellular states to better assess predictive models.

**Strengths:**

- The topic is relevant, as the prediction of genetic perturbations has critical biological applications and therapeutic potential, and previous work has shown how scFMs generally provide limited benefits in this context.
- The manuscript is easy to follow, and the figures are clear.
- A good number of scFMs is considered in this study.
- The work introduces a general, open-source benchmarking framework.

**Weaknesses:**

- Setting: the major limitations of this work are its scope and predictive methodology. This work focuses on a very specific setting of "zero-shot" scFM embeddings for perturbation effect prediction. However, this is not necessarily the only (or the best) approach to leverage scFMs for this task. Notably, previous works on scFMs also include perturbation effect prediction results and often focus on a different setting. For example, scGPT, in the original publication (Cui et al., 2024), leverages finetuning for this task, reporting promising results. This leads to important limitations: 1) while it is possible to compare multiple methods in this study, it is not possible to compare results to those included in the original publications, and 2) given that this is the only approach tested in this paper for all methods, it is difficult to say if it is generally good enough for the task. Indeed, it could be possible that the way perturbed embeddings are generated (line 133) leads to out-of-distribution (and, in turn, low-quality) scFMs embeddings for all methods. Overall, this limits the generality of the conclusions of this work.
- Novelty: the novelty of this work appears to be limited, as it leverages previously-defined evaluation metrics, and both the dataset, task, and models have been extensively studied in previous works.
- Data: the benchmark relies on a single, small dataset. This further restricts the generality of the conclusions, potentially limiting their robustness.


Overall, this work presents a general framework that, while potentially relevant, still has critical limitations in terms of modeling techniques explored and datasets analyzed.

**Questions:**

- The authors should include additional datasets, discussing trends across them.
- The authors should include multiple modeling paradigms, for example based on fine-tuning or exploring the space of different architectures for linear probing.
- The authors should include additional baselines, in particular specialized methods for perturbation effect prediction (e.g., GEARS), to better contextualize the results.

---

> ### Author Response · Authors · 2024-11-21
> **Datasets**
>
> We thank the reviewer for their thoughtful feedback and detailed comments on our work. We appreciate the recognition of the relevance of genetic perturbation prediction, the inclusion of multiple scFMs, the clarity of our manuscript and figures, and the open-source nature of our benchmarking framework. Below, we address the reviewer’s concerns in detail:
>
> **Datasets**
>
> Regarding the use of additional datasets, we agree this would broaden the scope of our analysis. We want to clarify that we originally focused on the Norman dataset because it has been shown to contain a higher-quality and stronger perturbation signal [See Figure 3.b. in [1]] and was therefore better suited to our analysis. We hypothesised that if scFM embeddings contained biologically relevant information for perturbation effect prediction, it would be more likely to manifest in a dataset with a strong signal like Norman. Positive results in this context would indicate that biologically relevant perturbation effects can be predicted accurately with zero-shot scFM embeddings, while negative results would provide greater confidence that the observed limitations reflect true negatives, rather than being due to low-quality signal inherent to the dataset tested.
>
> In line with this argument, the Replogle dataset, despite containing a greater number of perturbations (~4,000 K562; ~200 RPE1), demonstrates a lower-quality perturbation signal overall [1]. This may be due in part to the complexity of its perturbation space: seeing more perturbations simultaneously may introduce signal that is diffuse and not conclusively tied to any single perturbation, resulting in increased noise. This characteristic makes it less well-suited for drawing strong conclusions about scFM embeddings in their current form. However, we agree with the reviewers that incorporating additional datasets like Replogle would strengthen the paper and increase the credibility of our conclusions. While compute and memory constraints currently pose challenges to processing the Replogle dataset, we are working on engineering solutions to enable this analysis. We are committed to reporting these results as soon as they are available.
>
> The reviewer raises an important point about the potential for our embedding generation approach (line 133) to influence the quality of embeddings. Our approach involves generating perturbed cell embeddings by setting the expression of the perturbed gene to zero in all cells where it is expressed, effectively simulating a perturbation in silico. This methodology is consistent with approaches used in prior scFM papers. We acknowledge that this approach might not fully capture the complexity of biological perturbations, but it provides a consistent framework for benchmarking scFMs. Future work could explore alternative embedding generation strategies tailored to specific biological contexts.
>
> [1] Peidli *et al.*, 2024. scPerturb: harmonized single-cell perturbation data. https://www.nature.com/articles/s41592-023-02144-y

---

> ### Author Response · Authors · 2024-11-21
> **Setting / modelling paradigm**
>
> **Setting / modelling paradigm**
>
> Thank you for the feedback on this aspect of our work. We would first like to note that even in the absence of fine-tuning, if a signal relative to perturbation effect prediction were present in the sc-embeddings, it should still be measurable in a zero-shot setting. If no performance improvement comes from foundation model embeddings and all observed performance gains are due to fine-tuning, this contradicts the guiding principle that these models learn useful biological representations during pre-training. MLP probes are commonly used as an approach to answer the question of information content of embeddings in NLP and CV, as they evaluate representation quality while removing confounding effects pertaining to task-specific prediction heads, which introduce inductive biases [2,3,4]. We also note that previous work has investigated the performance of fine-tuned versions of scGPT and scFoundation [5]. This study found that simple linear baselines still outperformed these models, suggesting that fine-tuning does not fully address their limitations. Because fine-tuning performance is addressed in other studies and because it fundamentally goes against our approach of establishing existing information content, we do not include fine-tuning in our study design. The findings we present therefore highlight that scFM embeddings have not learnt useful biological patterns pertaining to the perturbation prediction task. This in and of itself is an important finding.
>
>
> [2] Radford et al., 2021. Learning Transferable Visual Models From Natural Language Supervision. https://arxiv.org/abs/2103.00020
>
> [3] Tenney et al., 2024. What do you learn from context? Probing for sentence structure in contextualized word representations https://openreview.net/forum?id=SJzSgnRcKX
>
> [4] Boiarsky et al., 2023. A Deep Dive into Single-Cell RNA Sequencing Foundation Models. https://www.biorxiv.org/content/10.1101/2023.10.19.563100v1
>
> [5] Ahlmann-Eltze et al., 2024. Deep learning-based predictions of gene perturbation effects do not yet outperform simple linear methods. https://www.biorxiv.org/content/10.1101/2024.09.16.613342v4

---

> ### Author Response · Authors · 2024-11-21
> **Novelty**
>
> **Novelty**
>
> While we acknowledge that some of the evaluation metrics and models included in our benchmark have been previously explored, PertEval-scFM introduces several key contributions that add value to the field:
>
> * **Systematic distribution shift evaluation:** The inclusion of the SPECTRA framework allows us to evaluate scFM robustness under varying train-test dissimilarity, a critical aspect for real-world applications that involve significant domain shifts.
> * **Comprehensive metrics**: By integrating AUSPC, E-distance, and contextual alignment, PertEval-scFM provides a more holistic view of model performance, focusing on robustness and sensitivity to distribution shifts rather than accuracy alone.
> * **Open-source framework**: Our codebase is designed to be extendable, enabling the community to incorporate new datasets, models, and metrics, fostering ongoing development in this area.
>
> These contributions aim to establish a standardised and reproducible evaluation framework for scFMs, addressing a gap in the field despite the previously studied task and models.

---

> ### Author Response · Authors · 2024-11-21
> **GEARS as an additional baseline**
>
> **Additional baseline**
>
> In Table 1 we include results where we have implemented GEARS on the raw expression, using SPECTRA splits. GEARS obtains strong predictive performance, outperforming baseline models, consistent with its purpose built architecture specifically designed for Perturbation effect prediction. Similarly to other models, GEARS achieves better performance in prediction of overall perturbation effect compared to effect on the top 20 differentially expressed genes (DEGs). This points to an inherent limitation in the ability to capture fine-grained perturbation-specific responses (such as in highly regulated DEGs). We again observe an increase in MSE with increasing sparsification probability, a pattern consistent with PertEval findings. This shows that increasing train-test dissimilarity reduces model robustness, emphasising the difficulty of generalising to out-of-distribution perturbations. This underscores the need for robust evaluation frameworks that take distribution shift into account and highlights both current limitations and future directions for research. In tandem with the finding scFM embeddings do not provide meaningful information for perturbation prediction, these results advocate for the development of specialised models for perturbation prediction tasks, as opposed to adapting general-purpose scFMs, which is one of the conclusions of our paper.
>
> **Table 1: MSE for perturbation effect prediction with GEARS trained on raw expression data across increasing sparsification probabilities**
>
> **Overall MSE**
> |        | 0.0          | 0.1          | 0.2          | 0.3          | 0.4          | 0.5          | 0.6          | 0.7          | AUSPC            | Δ AUSPC |
> |--------|--------------|--------------|--------------|--------------|--------------|--------------|--------------|--------------|------------------|---------|
> | MSE ± SEM | 0.005500 ± 0.000231 | 0.008867 ± 0.002022 | 0.009367 ± 0.001770 | 0.011200 ± 0.001670 | 0.016933 ± 0.003284 | 0.017500 ± 0.004013 | 0.010067 ± 0.004270 | 0.010000 ± 0.002570 | 0.008148 ± 0.000393 |     0.037968    |
>
> **Top 20 DEGs**
> |        | 0.0          | 0.1          | 0.2          | 0.3          | 0.4          | 0.5          | 0.6          | 0.7          | AUSPC            | Δ AUSPC |
> |--------|--------------|--------------|--------------|--------------|--------------|--------------|--------------|--------------|------------------|---------|
> | MSE ± SEM | 0.239933 ± 0.024731 | 0.284067 ± 0.024137 | 0.215367 ± 0.037126 | 0.313867 ± 0.070730 | 0.255533 ± 0.034652 | 0.340533 ± 0.014432 | 0.681700 ± 0.194295 | 0.888233 ± 0.285305 | 0.265515 ± 0.017898 |    0.079672     |

---

> ### Author Response · Authors · 2024-11-21
> **Conclusion**
>
> We thank the reviewer for their constructive feedback, which has highlighted important areas for improvement. We believe that our focus on zero-shot evaluation, systematic distribution shift analysis, and comprehensive metrics provides significant contributions to the field, while our ongoing efforts to include additional datasets and advanced baselines will address the limitations raised. By continuing to expand the PertEval-scFM framework and its accompanying GitHub resource, we provide a valuable and evolving benchmarking tool for the community. Thank you again for your detailed comments.

---

> > ### Comment · Reviewer_SABf · 2024-11-27
> >
> > I would like to thank the authors for the response and the additional details. I appreciated the constructive comments and additional details.
> >
> >
> > **Dataset**
> >
> > Thank you for the additional details. Adding new datasets would greatly improve the robustness of the current findings. I checked the current version of the PDF and could not find the Replogle results noted in the reply, more details on that wuld be appreciated.
> >
> > **Setting / modelling paradigm**
> >
> > In my opinion this remains the main limitation of this work. The cited works definitely support a correlation between linear probing and finetuning, but the same works also found that these are not necessarily equivalent, only correlated, with significant changes between the two metrics. Additionally, the models are different (for example, based on contrastive learning).
> > Other works investigated the relationship between finetuning and linear probing deeply (e.g., Kumar et al., ICLR 2022; Zhu et al., EMNLP 2022), and such relationship, although overall correlated, appears to be complex, model- and task-specific.
> >
> > However, to further clarify,  I am not specifically concerned about the usage of linear probing by itself.
> > My main concern is that the *only* approach used in this paper is based on a novel technique in this domain which 1) is different from what is reported in the original publications and 2) is based on strong assumptions, such as the way perturbed embeddings are generated (line 133).
> > At this point, given that *all* the tested methods report mediocre results, there are two options: either all the methods do not capture relevant biological knowledge, or the setting of this benchmark is not able to leverage it.
> > This is a crucial (and very open) question that this manuscript does not help solve, directly jumping to the conclusion that existing methods are limited. Therefore, I believe that testing different prediction settings would at least help address this question. For example, if four different predictive methods (including those proposed in the original publications, such as finetuning) all report mediocre performance, I would be more inclined to support the conclusion that existing methods are indeed limited.
> >
> > Regarding Ahlmann-Eltze et al. 2024, I agree that this is a very related study but, again, the different settings make the results hard to compare.
> >
> > **Novelty**
> >
> >
> > Thank you for further clarifying this point. Even though I still believe that the novelty is somehow limited, I agree that combining all these aspects would make the overall contribution interesting, once the other points raised are solved.
> >
> > **Additional baselines**
> >
> > Thank you for integrating GEARS. I think that adding this kind of baseline improves the robustness of the approach.
> > However, it also creates more questions; for example: how does it compare to scGPT and other models benchmarked in the paper? (the table you included does not specify the setting, so it is hard to compare).
> >
> >
> > Given that my main concern (method setting) is still unsolved, and the other points (additional datasets, additional baselines) are also not fully clear, I decided to keep my score.

---

> > > ### Author Response · Authors · 2024-11-29
> > >
> > > We sincerely thank Reviewer SABf for the detailed feedback and for acknowledging the constructive updates made to our work. Below, we address the concerns raised point-by-point:
> > >
> > > **Dataset**
> > > We have made significant progress in incorporating the Replogle et al. (2022) dataset, with preliminary results included in the revised submission, and we anticipate completing the full set of results within approximately one week for inclusion in the final manuscript.
> > >
> > > **Setting / Modelling Paradigm**
> > > We appreciate the reviewer’s insightful comments regarding the distinction between linear probing and fine-tuning, as explored in works like Kumar et al. (2022) and Zhu et al. (2022). The nuanced relationship between these approaches is indeed relevant to this discussion. The reviewer rightfully notes:
> > >
> > > >“At this point, given that all the tested methods report mediocre results, there are two options: either all the methods do not capture relevant biological knowledge, or the setting of this benchmark is not able to leverage it. This is a crucial (and very open) question that this manuscript does not help solve, directly jumping to the conclusion that existing methods are limited. Therefore, I believe that testing different prediction settings would at least help address this question.”
> > >
> > > We agree that fine-tuning could provide valuable insights into whether the benchmark settings can better leverage the information within scFM embeddings. However, we emphasise that the focus of this study is on assessing the zero-shot information content of these embeddings, which is critical for understanding their inherent biological relevance. To complement this zero-shot evaluation, we included results from the GEARS baseline. GEARS convincingly outperforms scFM embeddings by incorporating inductive biases such as graph-based Gene Ontology embeddings to represent perturbations. This result highlights that scFM embeddings alone lack the perturbation-specific information necessary for accurate predictions. Instead, the success of GEARS suggests that additional perturbation-specific inductive biases are essential, which we believe that current embedding paradigms do not adequately capture.
> > >
> > > This outcome underscores a key limitation in the design of existing foundation models. We believe that a valuable direction for future research would involve designing foundation models that incorporate perturbation-specific inductive biases, as we suggest in our manuscript. Models like scFoundation \[2\] and recent concurrent work by *Liu et al.* (2024) \[3\] provide early evidence in this direction. Moreover, Liu et al. demonstrate that for scGPT, pre-training contributes minimally to perturbation prediction performance, with most of the predictive power attributed to the decoder head (Supp. Mat. Ext. Fig. 22). They also emphasized the need for a more detailed investigation into the zero-shot capabilities of scFM embeddings, a gap that our work specifically addresses.
> > >
> > > We maintain that our zero-shot framework provides a biologically meaningful lens for interpreting scFM embeddings, minimising the confounding effects introduced by fine-tuning or task-specific inductive biases. While fine-tuning remains an important extension for future exploration, we believe our zero-shot focus is both timely and necessary for evaluating the current generation of scFMs.
> > >
> > > **Additional Baselines**
> > >
> > > We included GEARS as a state-of-the-art baseline for perturbation prediction, faithfully reproducing its original implementation while aligning train-test splits with the SPECTRA framework, ensuring consistency and enabling robust comparisons; details are available in the revised manuscript.
> > >
> > > **Summary**
> > > In conclusion, we are confident in the significance of our zero-shot evaluation framework and the meaningful insights it provides into the biological relevance of scFM embeddings for perturbation prediction. While we acknowledge and value the suggestions for extending the scope of our study, we believe that our current contributions are impactful. They shed light on the strengths and limitations of scFM representations, their applicability in zero-shot perturbation prediction, observed failure modes, and offer well-supported recommendations for guiding the development of future iterations of such models.
> > >
> > > Thank you for your thoughtful feedback and constructive comments.
> > >
> > > \[1\] Replogle, J. M. *et al.* Mapping information-rich genotype–phenotype landscapes with genome-scale Perturb-seq. Cell 185, 2559–2575 (2022).
> > >
> > > \[2\] Hao, M., Gong, J., Zeng, X. *et al.* Large-scale foundation model on single-cell transcriptomics. *Nat Methods* 21, 1481–1491 (2024).
> > >
> > > \[3\] Liu, T. *et al.* Evaluating the Utilities of Foundation Models in Single-cell Data Analysis. bioRxiv doi: [https://doi.org/10.1101/2023.09.08.555192](https://doi.org/10.1101/2023.09.08.555192) (2024)

---

### Official Review · Reviewer_JL1w · 2024-11-04

**Soundness:** 3
**Presentation:** 3
**Contribution:** 2
**Rating:** 6
**Confidence:** 4

**Summary:**

This work presents PertEval-scFM for evaluating zero-shot single-cell foundation models (scFMs) in predicting cellular responses to genetic perturbations, using the Norman et al. dataset (2019).  Specifically, PertEval-scFM evaluates zero-shot embeddings from several existing scFMs—such as scGPT, scBERT, Geneformer, UCE, and scFoundation against baseline models based on gene expression matrix in the distribution shift scenario. The results suggest that zero-shot scFM embeddings do not consistently outperform simpler baselines, such as mean expression and MLP baselines across different evaluation metrics. As the sparsification probability increases, the performance of scFM embeddings declines more sharply than baseline models, suggesting that scFMs may be sensitive to distribution shifts. particularly in strong/atypical scenarios, with limited generalizability in zero-shot contexts of perturbation effect prediction.

**Strengths:**

**More scFMs in benchmarking and distribution shift scenarios**: Compared to existing works on gene perturbation benchmarking,  PertEval-scFM considers more available updated scFMs and introduces distribution shift scenarios to evaluate the generalization of scFM better.

**Comprehensive Metrics**: By combining AUSPC, E-distance, and contextual alignment,  PertEval-scFM provides a detailed analysis of model robustness, distribution shift sensitivity, and alignment between pre-training and application-specific datasets.

**Reproducibility**: The PertEval-scFM code has been provided transparently and accessiblely, with the potential for further development and use by the community.

**Weaknesses:**

**Novelty and Significance**: As the authors have noted in the introduction section, benchmarking scFMs on genetic perturbation tasks is not new, with similar conclusions being drawn.  The current benchmark indeed falls short in considering only one Perturb-seq dataset, making the results here rather preliminary. Using the datasets provided in existing benchmark works, such as Wu et al. (2024) could enhance the benchmarking breadth of current work. The author may also elaborate on addressing the key findings of PertEval-scFM by introducing distribution shift, especially in a biological context -- for example,  more quantitative and systematic analysis on the strong/atypical perturbations, and biological mechanism explanations/insights evidenced by bioinformatics analysis.

**The MLP Structure Selection**:  In PertEval-scFM， one hidden layer structure was used as an MLP probe to model the perturbation effects, and it served as a baseline. It's not clear how this selection influences the results since it's possible that the mapping between scFM embeddings and perturbation is highly complicated. Robustness tests could be added here. It is also insightful to investigate the role of scFM embeddings in more advanced or state-of-the-art deep learning-based prediction models such as GEARS[1] or CELLOT[2].

**Absence of Fine-Tuning Component**: While the zero-shot test provides insights into scFM performance, real-world applications frequently involve fine-tuning to adapt models. Without a fine-tuning component, the evaluation lacks a practical measure of scFM potential in real-world perturbation prediction settings, especially where significant distribution shifts are possible. Incorporating fine-tuning into the analysis, or at least comparing zero-shot performance to fine-tuned baselines, would yield a more comprehensive understanding of scFM effectiveness in applied perturbation prediction.

Overall, while the paper is not recommended to be accpeted by ICLR at current form, I would like to increase the score if the concerns are adequately addressed by the authors.

[1]Roohani, Y., Huang, K. & Leskovec, J. Predicting transcriptional outcomes of novel multigene perturbations with GEARS. Nat Biotechnol 42, 927–935 (2024). https://doi.org/10.1038/s41587-023-01905-6

[2]Bunne, C., Stark, S.G., Gut, G. et al. Learning single-cell perturbation responses using neural optimal transport. Nat Methods 20, 1759–1768 (2023). https://doi.org/10.1038/s41592-023-01969-x

**Questions:**

See weaknesses

---

> ### Author Response · Authors · 2024-11-21
> **Novelty and significance**
>
> We thank the reviewer for their feedback and constructive comments, as well as for acknowledging the strengths of our work, including the breadth of scFMs benchmarked, the inclusion of distribution shift scenarios, the comprehensive evaluation metrics, and the transparency and reproducibility of PertEval-scFM. Below, we address the reviewer’s concerns in detail:
>
> **Novelty and significance**
>
> We acknowledge that benchmarking scFMs on perturbation prediction tasks is not entirely novel. However, our work introduces unique contributions that differentiate PertEval-scFM:
> Distribution shift evaluation: we introduce systematic sparsification of train-test overlaps through the SPECTRA framework, providing a quantitative measure of how scFM performance is affected by increasing train-test dissimilarity. This approach offers insights into the robustness of scFM embeddings under realistic out-of-distribution scenarios, which are highly relevant to biological applications.
> Comprehensive metrics: the combination of AUSPC, E-distance, and contextual alignment provides a more detailed analysis of model performance than previous benchmarks. While existing studies have focused on accuracy-based metrics, we focus on robustness and sensitivity to distribution shifts, which are critical for real-world applications of scFMs.
> Regarding the use of additional datasets such as those in Wu et al. (2024), we agree this would broaden the scope of our analysis. We want to clarify that we originally focused on the Norman dataset because it has been shown to contain a higher-quality and stronger perturbation signal [See Figure 3.b. in [1]] and was therefore better suited to our analysis. We hypothesised that if scFM embeddings contained biologically relevant information for perturbation effect prediction, it would be more likely to manifest in a dataset with a strong signal like Norman. Positive results in this context would indicate that biologically relevant perturbation effects can be predicted accurately with zero-shot scFM embeddings, while negative results would provide greater confidence that the observed limitations reflect true negatives, rather than being due to low-quality signal inherent to the dataset tested.
>
> In line with this argument, the Replogle dataset, despite containing a greater number of perturbations (~4,000 K562; ~200 RPE1), demonstrates a lower-quality perturbation signal overall [1]. This characteristic makes it less well-suited for drawing strong conclusions about scFM embeddings in their current form. However, we agree with the reviewers that incorporating additional datasets like Replogle would strengthen the paper and increase the credibility of our conclusions. While compute and memory constraints currently pose challenges to processing the Replogle dataset, we are working on engineering solutions to enable this analysis. We aim to report these results as soon as they are available.
> We appreciate the reviewer’s suggestion to provide a more quantitative and systematic analysis of strong/atypical perturbations and their biological context. To address this, we performed additional correlation analyses to investigate the relationship between the perturbation effect size, as measured by energy distance (E-distance), and model performance (MSE). In the updated Figure 3 (see the attached pdf), we have included the Spearman correlation coefficient to quantify these observed relationships and added two-gene perturbations to increase the statistical power of the analysis. The results indicate that prediction performance does degrade as E-distance increases, but the relationship is weak ($\text{Spearman’s R} = 0.13, \text{p} = 1.22 × 10^{⁻⁸}$).  This suggests that the model’s capability to predict perturbation effects depends not only on the magnitude of the perturbation, but also on its distribution. For example, we observed that some pronounced perturbations, such as CEBPA, are well-predicted despite having a high E-distance. This can be explained by the narrow, pronounced peak of the effect (which inflates the E-distance) combined with a long tail of minimal effects, which makes the overall distribution easier to predict. This shows how we can use the E-distance to pinpoint interesting perturbations and failure modes beyond perturbation strength. This analysis reflects the role of PertEval-scFM as a benchmarking tool: the metrics we include (e.g. E-distance and AUSPC) are designed to enable model meaningful evaluation from a methodological as well as a biological perspective. They complement standard bioinformatic analyses approaches, by further connecting model predictions to meaningful biological mechanisms.
>
> [1] https://www.nature.com/articles/s41592-023-02144-y

---

> ### Author Response · Authors · 2024-11-21
> **MLP structure selection**
>
> **MLP structure selection**
>
> We acknowledge the reviewer’s concern regarding the single-layer MLP probe and its potential limitations. However, our choice of MLP architecture was motivated by its simplicity, which allows for isolating the information content of scFM embeddings without introducing additional confounding factors from complex prediction heads. Indeed, one of the main aims of our work was to evaluate the representation quality of the scFM embeddings while minimising the amount of inductive bias introduced by model architecture. A simple, one-layer MLP structure has been used in other comparable studies [2,3]. This approach also allows us to evaluate representation robustness under distribution shift, without confounding results via task-specific prediction head inductive bias [4].
>
> However, we also conducted experiments with varying MLP structures and found that increasing the complexity of the MLP had no significant impact on the results. We trained a range of MLPs with increasing parameter count on our raw expression data to verify the effect of parameter number on results. We report our findings in Table 1, where it can be seen the increase in parameters has no effect on the MSEs obtained. We also include additional results using scBERT and scFoundation embeddings as input, with increasing parameter count. This further strengthens our conclusion that, despite an increase in the expressiveness of the underlying MLP model, the scFM embeddings do not contain sufficient biologically relevant information from the pre-training stage to provide increased performance for this task.
>
>
> **Table 1: Train and Test set results with MLPs of increasing parameter count**
>
> | Trainable parameters (million) | train/MSE   | val/MSE   | Training data             |
> |--------------------------------|-------------|-----------|-------------------|
> | 1.6                            | 0.057067    | 0.057642  | Raw gene expression |
> | 3.2                            | 0.058670    | 0.057493  | Raw gene expression |
> | 6.3                            | 0.056748    | 0.057424  | Raw gene expression |
> | 12.7                           | 0.056724    | 0.057428  | Raw gene expression |
> | 1.6                            | 0.060780    | 0.060260  | scFoundation embeddings     |
> | 3.2                            | 0.060440    | 0.059910  | scFoundation embeddings    |
> | 12.6                           | 0.059570    | 0.059050  | scFoundation embeddings      |
> | 0.2                            | 0.061040    | 0.061426  | scBERT embeddings            |
> | 1                              | 0.061046    | 0.061428  | scBERT embeddings            |
> | 8                              | 0.061040    | 0.061421  | scBERT embeddings            |
>
> Despite the disparities in parameter count, our current approach maintains alignment with the standard practice of tailoring MLP architectures to the embedding size to ensure sufficient expressiveness for each model. This guarantees that no embedding is artificially constrained or unfairly advantaged due to mismatched architecture capacity.
>
> [2] Goyal *et al.*, 2023. Finetune Like You Pretrain: Improved Finetuning of Zero-Shot Vision Models https://openaccess.thecvf.com/content/CVPR2023/html/Goyal_Finetune_Like_You_Pretrain_Improved_Finetuning_of_Zero-Shot_Vision_Models_CVPR_2023_paper.html
>
> [3] Radford *et al.*, 2021 Learning Transferable Visual Models From Natural Language Supervision https://arxiv.org/abs/2103.00020
>
> [4] Heim *et al.* 2023, Towards Better Understanding of Domain Shift on Linear-Probed Visual Foundation Models https://openreview.net/forum?id=l188N6IZNY

---

> ### Author Response · Authors · 2024-11-21
> **scFM embeddings in GEARS and CELLOT**
>
> **scFM embeddings in GEARS and CELLOT**
>
> With regards to including results on scFM embeddings with GEARS and CellOT, we note that for GEARS we can only integrate gene embeddings per design, as GEARS explicitly uses a gene coexpression knowledge graph and Gene Ontology (GO)-derived relationships to learn embeddings that represent gene-specific perturbation effects. Cell embeddings, which encapsulate whole-cell states, would not align directly with these encoders unless significant architectural changes were made. This would involve revising the core design of GEARS to account for multiscale modelling, which we believe is beyond the scope of our paper. CellOT is designed to learn mappings between unpaired distributions of untreated and treated cells via Optimal Transport. In contrast, embeddings generated by scFM should capture intrinsic properties of cells without specific perturbation context. Integrating embeddings from zero-shot single-cell foundation models as inputs to CellOT will likely not align with CellOT's design, which requires raw single-cell measurements from both untreated and perturbed states to learn accurate mappings and embeddings. Therefore we believe it’s not appropriate for us to incorporate these prediction models within our benchmark.

---

> ### Author Response · Authors · 2024-11-21
> **Absence of fine-tuning component**
>
> **Absence of fine-tuning component**
>
> We understand the reviewer's concern regarding the fact that we are not fine-tuning the scFMs. We would first like to note that even in the absence of fine-tuning, if a signal relative to perturbation effect prediction were present in the sc-embeddings, it should still be measurable in a zero-shot setting. If no performance improvement comes from foundation model embeddings and all observed performance gains are due to fine-tuning, this contradicts the guiding principle that these models learn useful biological representations during pre-training. MLP probes are commonly used as an approach to answer the question of information content of embeddings in NLP and CV, as they evaluate representation quality while removing confounding effects pertaining to task-specific prediction heads, which introduce inductive biases [3, 5, 6]. We also note that previous work has investigated the performance of fine-tuned versions of scGPT and scFoundation [7]. This study found that simple linear baselines still outperformed these models, suggesting that fine-tuning does not fully address their limitations. Because fine-tuning performance is addressed in other studies and because it fundamentally goes against our approach of establishing existing information content, we do not include fine-tuning in our study design. The findings we present therefore highlight that scFM embeddings have not learnt useful biological patterns pertaining to the perturbation prediction task. This in and of itself is an important finding.
>
> [3] Radford *et al.*, 2021. Learning Transferable Visual Models From Natural Language Supervision. https://arxiv.org/abs/2103.00020
>
> [5] Tenney *et al.*, 2024. What do you learn from context? Probing for sentence structure in contextualized word representations https://openreview.net/forum?id=SJzSgnRcKX
>
> [6] Boiarsky *et al.*, 2023. A Deep Dive into Single-Cell RNA Sequencing Foundation Models.
> https://www.biorxiv.org/content/10.1101/2023.10.19.563100v1
>
> [7] Ahlmann-Eltze *et al.*, 2024. Deep learning-based predictions of gene perturbation effects do not yet outperform simple linear methods. https://www.biorxiv.org/content/10.1101/2024.09.16.613342v4

---

> ### Author Response · Authors · 2024-11-21
> **Conclusion**
>
> We appreciate the reviewer’s thoughtful feedback and the opportunity to improve our work. PertEval-scFM provides significant contributions in robust evaluation under distribution shifts, comprehensive metrics, and transparent benchmarking, all aimed at advancing scFM research. We plan to expand our framework to include additional datasets like Replogle, provide more biological insights into atypical perturbations, and explore the integration of scFM embeddings into advanced predictive models. These updates, along with continuous expansion of our accompanying GitHub resource, will ensure that PertEval-scFM remains a valuable and evolving resource for the community. Thank you again for your constructive feedback.

---

> > ### Comment · Reviewer_JL1w · 2024-11-27
> >
> > Thank you for your detailed response and the additional experiments. Although my concerns have not been fully resolved, considering the authors’ hard work I am willing to increase my score by one point.

---

> ### Comment · Reviewer_JL1w · 2024-11-27
>
> As stated in my previous comments, I expected to see more datasets, and scenarios (including fine-tuning) in the current work, which would make it a significant contribution to the community. Unfortunately, they seemed not to be present in the revision, nor in the author's immediate action plan, which prevented me from giving a higher score.

---

> > ### Author Response · Authors · 2024-11-29
> >
> > We sincerely thank the reviewer for acknowledging our hard work and for raising their score. We greatly appreciate the detailed feedback and the opportunity to further clarify and address the concerns raised.
> >
> > We hope that by incorporating the additional baseline (GEARS), extending our analysis to combinatorial perturbations, and making significant progress on the Replogle dataset \- as detailed in the general rebuttal and revised manuscript \- we have addressed your concerns.
> >
> > On the topic of **fine-tuning**, we maintain the original focus of our work: assessing the zero-shot information content of scFM embeddings for perturbation prediction. While fine-tuning is undoubtedly a valuable direction, the primary aim of this paper is to establish a standardized framework for evaluating biologically meaningful perturbation effects in a zero-shot setting, thereby probing the inherent information content of the embeddings. Extending our framework to incorporate fine-tuning poses several challenges that are beyond the scope of this work. These include:
> >
> > 1. *Computational resources*:  fine-tuning requires extensive computational and financial resources which we do not currently have at our disposal \- and indeed which many people do not have, highlighting the importance of this framework and these results for the research community;
> > 2. *Model-specific constraints*: some necessary components (e.g. prediction head for perturbation effect) are absent in certain foundation models, which would hinder fair and consistent comparisons;
> > 3. *Integration complexity*: Incorporating fine-tuning into the framework would require significant methodological adaptations, which we believe merit separate investigation.
> >
> > That said, we view our current framework as complementary to fine-tuning efforts, providing an essential foundation for understanding scFM embeddings in a zero-shot context. Future iterations of this work could build on this foundation to explore fine-tuning setups more comprehensively.
> >
> > We hope this additional context and clarification address the reviewer’s concerns. We remain committed to advancing this work and appreciate your thoughtful feedback.

---

### Author Response · Authors · 2024-11-21
**General rebuttal**

**General Introduction**

We would first like to thank all reviewers for their time, as well as for their insights and comments on our work. We note all reviewers valued the scope of single-cell Foundation Models (scFM) evaluated in our Benchmark PertEval-scFM, corresponding to most currently available models (JL1w, SABf, 3E1k). The suite of comprehensive metrics was also noted as a strength: we incorporate AUSPC, E-distance and contextual alignment into our framework, with the aim of providing standardised evaluation protocols for a rapidly expanding field (JL1w, 3E1k). In particular, we highlight the importance of quantifying model robustness and sensitivity to distribution shift in a field with critical biological/therapeutic applications (JL1w, 3E1k). This highlights a crucial yet often overlooked aspect of machine learning evaluations: the importance of generalizability across diverse and unseen scenarios. Integrating the AUSPC and E-distance to our pipeline to quantify these aspects has, to our knowledge, not been proposed elsewhere to date (3E1k, yfBT). Finally, we are grateful that the reproducibility and transparency of our work is recognised by the reviewers (JL1w, SABf, 3E1k, yfBT), with the code base and single-cell embeddings of our benchmark being maintained and added to as a resource for our research community.

**Aim**

The aim of this paper is to evaluate single-cell foundation models in a zero-shot context. This is because we want to quantify the information content of the single-cell embeddings in the context of perturbation prediction. The hypothesis is that due to the extensive pre-training on single-cell (sc) data these models have undergone, the embeddings obtained should contain biologically relevant information. Perturbation effect prediction is a major biological problem and knowing whether the models currently available are learning inherent biological principles pertaining to this task is an important research consideration. Identifying and understanding any limitations will guide the development of better models and help advance the field.

**Concerns**

In each following comment, we address the concerns which were raised by all reviewers and respond to individual concerns as comments to the corresponding reviews:
1. Fine-tuning
2. Additional Baseline
3. MLP parameter count
4. Dataset
5. SPECTRA and Dataset Splitting

---

> ### Author Response · Authors · 2024-11-21
> **Fine-tuning**
>
> **Fine-tuning**
>
> One of the concerns raised by reviewers (**JL1w**, **SABf**) was that we were not fine-tuning the scFMs in this paper. We would first like to note that even in the absence of fine-tuning, if a signal relative to perturbation effect prediction were present in the sc-embeddings, it should still be measurable in a zero-shot setting. If no performance improvement comes from foundation model embeddings and all observed performance gains are due to fine-tuning, this contradicts the guiding principle that these models learn useful biological representations during pre-training. MLP probes are commonly used as an approach to answer the question of information content of embeddings in NLP and CV, as they evaluate representation quality while removing confounding effects pertaining to task-specific prediction heads, which introduce inductive biases [1,2,3]. We also note that previous work has investigated the performance of fine-tuned versions of scGPT and scFoundation [4]. This study found that simple linear baselines still outperformed these models, suggesting that fine-tuning does not fully address their limitations. Because fine-tuning performance is addressed in other studies and because it fundamentally goes against our approach of establishing existing information content, we do not include fine-tuning in our study design. The findings we present therefore highlight that scFM embeddings have not learnt useful biological patterns pertaining to the perturbation prediction task. This in and of itself is an important finding.
>
> [1] https://openreview.net/forum?id=SJzSgnRcKX
>
> [2] https://arxiv.org/abs/2103.00020
>
> [3] https://www.biorxiv.org/content/10.1101/2023.10.19.563100v1
>
> [4] https://www.biorxiv.org/content/10.1101/2024.09.16.613342v1

---

> ### Author Response · Authors · 2024-11-21
> **Additional Baseline**
>
> **Additional Baseline**
>
> To answer reviewers' concerns about including additional baselines (**SABf**), in Table 1 we include results where we have implemented GEARS on the raw expression, using SPECTRA splits. GEARS achieves strong predictive performance, outperforming baseline models, consistent with its purpose-built architecture specifically designed for perturbation effect prediction. Similarly to other models, GEARS achieves better performance in prediction of the overall perturbation effect compared to the effect on the top 20 differentially expressed genes (DEGs). This is in line with our previous observation that in general, fine-grained perturbation responses are harder to predict. We again observe an increase in MSE with increasing sparsification probability, a pattern consistent with our other findings. This shows that even highly specific perturbation prediction models suffer under increasing train-test dissimilarity, emphasising the difficulty of generalising to out-of-distribution perturbations. This underscores the need for robust evaluation frameworks that take distribution shift into account to assess model generalisability and highlight both current limitations and future directions for research. Alongside the finding that zero-shot scFMs fail to provide meaningful representations for perturbation prediction, these results suggest that, instead of focusing on building general-purpose scFMs, the successful prediction of perturbation effects requires incorporating perturbation-specific design principles into the architecture of scFMs.
>
> **Table 1: MSE for perturbation effect prediction with GEARS trained on raw expression data across increasing sparsification probabilities**
>
> **Overall MSE**
>
> |        | 0.0          | 0.1          | 0.2          | 0.3          | 0.4          | 0.5          | 0.6          | 0.7          | AUSPC            | Δ AUSPC |
> |--------|--------------|--------------|--------------|--------------|--------------|--------------|--------------|--------------|------------------|---------|
> | MSE ± SEM | 0.005500 ± 0.000231 | 0.008867 ± 0.002022 | 0.009367 ± 0.001770 | 0.011200 ± 0.001670 | 0.016933 ± 0.003284 | 0.017500 ± 0.004013 | 0.010067 ± 0.004270 | 0.010000 ± 0.002570 | 0.008148 ± 0.000393 |     0.037968    |
>
> **Top 20 DEGs**
>
> |        | 0.0          | 0.1          | 0.2          | 0.3          | 0.4          | 0.5          | 0.6          | 0.7          | AUSPC            | Δ AUSPC |
> |--------|--------------|--------------|--------------|--------------|--------------|--------------|--------------|--------------|------------------|---------|
> | MSE ± SEM | 0.239933 ± 0.024731 | 0.284067 ± 0.024137 | 0.215367 ± 0.037126 | 0.313867 ± 0.070730 | 0.255533 ± 0.034652 | 0.340533 ± 0.014432 | 0.681700 ± 0.194295 | 0.888233 ± 0.285305 | 0.265515 ± 0.017898 |    0.079672     |

---

> ### Author Response · Authors · 2024-11-21
> **MLP parameter count**
>
> **MLP parameter count**
>
> Another concern raised (**JL1w**, **3E1k**) is the discrepancy in the number of parameters of the MLPs used to probe the scFM embeddings, given that these embeddings have a variable size which is a parameter of their corresponding scFM. To alleviate this concern, we trained a range of MLPs with increasing parameter count on our raw expression data to verify the effect of parameter number on results. We report our findings in Table 2, where it can be seen the increase in parameters has no effect on the MSEs obtained. We also include additional results using scBERT and scFoundation embeddings as input, with increasing parameter count. This further strengthens our conclusion that, despite an increase in the expressiveness of the underlying MLP model, the scFM embeddings do not contain sufficient biologically relevant information from the pre-training stage to provide increased performance for this task. Despite the disparities in parameter count, our current approach maintains alignment with the standard practice of tailoring MLP architectures to the embedding size to ensure sufficient expressiveness for each model. This guarantees that no embedding is artificially constrained or unfairly advantaged due to mismatched architecture capacity.
>
> **Table 2: Train and Test set results with MLPs of increasing parameter count**
>
> | Trainable parameters (million) | Training data             | train/MSE   | val/MSE   |
> |--------------------------------|---------------------------|-------------|-----------|
> | 1.6                            | Raw gene expression       | 0.057067    | 0.057642  |
> | 3.2                            | Raw gene expression       | 0.058670    | 0.057493  |
> | 6.3                            | Raw gene expression       | 0.056748    | 0.057424  |
> | 12.7                           | Raw gene expression       | 0.056724    | 0.057428  |
> | 1.6                            | scFoundation embeddings   | 0.060780    | 0.060260  |
> | 3.2                            | scFoundation embeddings   | 0.060440    | 0.059910  |
> | 12.6                           | scFoundation embeddings   | 0.059570    | 0.059050  |
> | 0.2                            | scBERT embeddings         | 0.061040    | 0.061426  |
> | 1                              | scBERT embeddings         | 0.061046    | 0.061428  |
> | 8                              | scBERT embeddings         | 0.061040    | 0.061421  |

---

> ### Author Response · Authors · 2024-11-21
> **Dataset**
>
> **Dataset**
>
> One of the main concerns raised by reviewers (**JL1w, SABf, 3E1k, yfBT**) is that our results were only obtained on single-gene perturbations from the Norman dataset. We want to clarify that we originally focused on the Norman dataset because it has been shown to contain the strongest perturbation effect signal for genetic perturbations [See Figure 3.b. in [5]] and is therefore best suited to our analysis. We hypothesised that if scFM embeddings contain biologically relevant information for perturbation effect prediction, it would be more likely to be contained in a dataset with a strong perturbation signal. Positive results in this context would indicate that biologically relevant perturbation effects can be predicted accurately with zero-shot scFM embeddings, while negative results would provide greater confidence that the observed limitations reflect true negatives, rather than being due to low-quality signal inherent to the dataset tested. In line with this argument, the Replogle dataset, despite containing a greater number of perturbations (~4,000 K562; ~200 RPE1), demonstrates much diminished perturbation effects signal overall [See Figure 3.b. in [5]]. This makes it less well-suited for drawing strong conclusions about scFM embeddings in their current form. We have expanded our analysis to include all the combinatorial perturbations in the Norman dataset, where originally we only considered single-gene perturbations. We present the updated results in Tables 3 and 4.
>
> **Table 3: Perturbation effect prediction evaluation across 2000 HVGs.**
> |Model|Perturbation strategy|SP0.0|SP0.1|SP0.2|SP0.3|SP0.4|SP0.5|SP0.6|SP0.7|AUSPC|ΔAUSPC|
> |-----|--------------------|-----|-----|-----|-----|-----|-----|-----|-----|-----|------|
> |Mean baseline|Dual-gene|5.337±0.093|5.257±0.102|5.910±0.255|5.722±0.401|6.644±0.167|7.674±0.962|6.071±0.772|5.201±0.594|4.255±0.073|-|
> |MLP gene expression|Dual-gene|5.337±0.094|5.261±0.100|5.913±0.255|5.728±0.402|6.635±0.161|7.675±0.953|6.050±0.763|5.198±0.593|4.253±0.073|0.002|
> |scFoundation|Dual-gene|5.675±0.106|5.564±0.051|6.173±0.196|6.050±0.462|6.755±0.279|7.944±1.186|6.382±0.876|5.238±0.578|4.432±0.467|-0.177|
> |UCE|Dual-gene|5.655±0.091|5.514±0.066|6.145±0.183|6.029±0.460|6.736±0.289|7.939±1.184|6.352±0.831|5.612±0.665|4.435±0.085|-0.180|
> |Geneformer|Dual-gene|5.654±0.091|5.514±0.067|6.145±0.182|6.029±0.458|6.742±0.287|7.937±1.187|7.246±0.707|5.179±0.630|4.503±0.081|-0.248|
> |scBERT|Dual-gene|5.655±0.092|5.515±0.067|6.159±0.196|6.022±0.465|6.757±0.281|7.999±1.240|6.493±0.736|7.110±0.579|4.533±0.081|-0.278|
> |scGPT|Dual-gene|5.654±0.091|5.515±0.067|6.153±0.189|6.023±0.464|6.766±0.287|8.272±1.377|8.826±0.182|14.906±2.154|5.184±0.132|-0.929|
>
> **Table 4: Perturbation effect prediction evaluation across the top 20 DEGs.**
>
> *Note: For two-gene perturbations split 0.5, not enough perturbations passed our quality control to properly define the split.*
> |Model|Perturbation strategy|SP0.0|SP0.1|SP0.2|SP0.3|SP0.4|SP0.5|SP0.6|SP0.7|↓AUSPC(10⁻²)|↑ΔAUSPC(10⁻²)|
> |-----|--------------------|-----|-----|-----|-----|-----|-----|-----|-----|-----------|-------------|
> |Mean baseline|Dual-gene|2.524±1.054|0.549±0.055|0.580±0.075|0.615±0.074|0.653±0.037|-|0.659±0.047|0.497±0.056|0.522±0.053|-|
> |MLP gene expression|Dual-gene|0.195±0.121|0.484±0.046|0.538±0.082|0.585±0.061|0.618±0.048|-|0.552±0.049|0.500±0.056|0.371±0.00|15.1|
> |Geneformer|Dual-gene|0.489±0.043|0.527±0.055|0.550±0.069|0.603±0.076|0.661±0.045|-|0.623±0.054|0.487±0.048|0.409±0.008|11.3|
> |UCE|Dual-gene|0.489±0.043|0.527±0.055|0.550±0.069|0.601±0.072|0.656±0.043|-|0.624±0.053|0.506±0.048|0.410±0.007|11.2|
> |scFoundation|Dual-gene|0.493±0.044|0.534±0.057|0.554±0.070|0.606±0.073|0.656±0.045|-|0.621±0.060|0.497±0.051|0.410±0.008|11.2|
> |scBERT|Dual-gene|0.490±0.043|0.528±0.056|0.550±0.069|0.596±0.071|0.661±0.041|-|0.622±0.049|0.681±0.086|0.418±0.008|10.4|
> |scGPT|Dual-gene|0.489±0.043|0.527±0.056|0.550±0.069|0.597±0.072|0.673±0.044|-|0.724±0.028|1.941±0.329|0.499±0.018|2.2|
>
> The relative performance with respect to the mean baseline increases between single and double genes, however, the baseline gene expression MLP still performs best. This aligns with our prior observations that generalisability decreases for perturbations that are strong or atypical as indicated by higher AUSPCs. While the additional results provide further evidence of the trends observed in our initial analysis, they also underline the inherent limitations of current scFM embeddings in addressing the complexities of dual-gene perturbations and emphasise the necessity of future efforts to improve their robustness and biological relevance. Furthermore, we are currently working on including the Replogle *et al.* dataset in our benchmark. While memory constraints currently pose challenges to processing this dataset, we are working on solutions and aim to report these results as soon as possible.
>
> [5] https://www.nature.com/articles/s41592-023-02144-y

---

> ### Author Response · Authors · 2024-11-21
> **SPECTRA and Dataset Splitting**
>
> **SPECTRA and Dataset Splitting**
>
> We appreciate the concern regarding the impact of dataset splitting (**yfBT**). However, we believe SPECTRA’s contribution extends beyond what random splits or current experimental designs offer. In real-world applications like Perturb-seq, unpredictable biological distribution shifts are common. Random splits are not inherently easier or harder; a random split could align with a sparsification	probability of 0.1 or 0.7, representing significantly different train-test dissimilarities. However, systematically quantifying this dissimilarity makes it possible to contextualise performance results and compare model robustness across different tasks or datasets. Random splits may therefore fail to capture these distribution shifts, often leading to overly optimistic assessments of model performance. SPECTRA addresses this by generating splits of increasing difficulty, starting with random splits and progressing to more challenging ones based on train-test similarity. This approach ensures that SPECTRA encompasses random splits but also evaluates a model’s robustness across a broader spectrum of biological variability. SPECTRA does not artificially inflate task difficulty but rather controls for the similarity between train and test split. This indeed increases test difficulty, but not artificially. While the train-test similarity control may lead to smaller datasets in harder splits, this is a reflection of the redundancy inherent to the dataset, not a limitation of SPECTRA. Moreover, SPECTRA is not just an evaluation tool but a valuable guide for experimental design. For example, in Perturb-seq, SPECTRA can reveal if a model generalises well within pathways but struggles across pathways, guiding researchers on which genes to test next. Random splits, by contrast, obscure such insights due to high train-test similarity. We argue that SPECTRA’s ability to highlight weaknesses in generalisability, simulate real-world distribution shifts, and inform experimental design makes it a high-impact addition to evaluation pipelines, especially for tasks like Perturb-seq where navigating distribution shifts is critical.

---

> ### Author Response · Authors · 2024-11-21
> **Conclusion**
>
> In conclusion, we thank the reviewers for their thorough evaluation and constructive feedback. Through this rebuttal, we have addressed several key concerns:
> 1. We have clarified our rationale for evaluating models in a zero-shot setting, emphasising that this approach specifically targets the assessment of learned biological representations during pre-training. The observed limitations in perturbation effect prediction suggest that current scFMs may not be learning the intended biological principles during pre-training, an important finding for the field.
> 2. We have expanded our evaluation to include:
>     - GEARS as an additional baseline, demonstrating that even purpose-built architectures face challenges with distribution shift
> Results on dual-gene perturbations, which further validate our initial findings while highlighting the increased difficulty of predicting complex perturbation effects
>    - Analysis of MLP parameter counts, confirming that our findings are robust to variations in model capacity
> 3. We have provided a detailed justification for our dataset selection, explaining that the Norman dataset's strong perturbation signal makes it particularly suitable for evaluating whether scFMs capture perturbation-related biological information. We are actively working to extend our analysis to the Replogle dataset to further strengthen our conclusions.
> 4. We have defended SPECTRA's value as an evaluation framework, emphasising its ability to systematically quantify distribution shifts and provide insights that random splits cannot capture. This approach is particularly relevant for real-world applications where biological variability poses significant challenges.
>
> Our findings suggest that instead of pursuing general-purpose scFMs, perturbation-specific design principles should be incorporated into scFM model architectures. We believe our benchmark makes a valuable contribution by providing standardised evaluation protocols that can guide such future developments. We have open-sourced these tools and the framework as a resource for the research community.
>
> We look forward to incorporating these clarifications and additional results into our revised manuscript. We believe these changes will strengthen our paper while maintaining its core contribution: a comprehensive evaluation framework that highlights both the current limitations and future opportunities in single-cell foundation models.

---

### Author Response · Authors · 2024-11-28

We sincerely thank the reviewers for acknowledging our hard work and for their detailed feedback, which has helped us enrich and expand our work. Taking on board all your comments, we have made substantial additions to the revised PDF, which we describe below:

**1 - Additional Baseline.** We have included **GEARS** as a **task-specific baseline**, enriching the contextual relevance of our comparisons. See **Section 2.2.2**, **Tables 2** and **3** and **Section 3.2** in the PDF.

**2 - Dual-gene perturbation.** We have extended our analysis to encompass not only single-gene perturbations but also **combinatorial perturbations (dual-gene)** from the Norman dataset. See **Section 2.1.2**, **Tables 2** and **3** and **Section 3.2** in the PDF. This is a non-trivial extension which provides a broader and more comprehensive evaluation of our framework.

**3 - Additional Dataset.** Crucially, we have incorporated the **Replogle et al. (2022) dataset** [1] to our analysis. See **Section 2.4.1** and **Tables 2** and **3** for results in the PDF. Preprocessing this dataset (~2,000 perturbations) presents a significant computational burden, which we have been working hard to overcome. All data preprocessing is complete, and we have developed a fully functional featurization pipeline. Preliminary results, spanning **three models and two baselines across five SPECTRA splits**, are included as a demonstration of our progress. We anticipate completing the full set of results within one week.

**4 - MLP parameter count.** We included an analysis on the effect of MLP parameter count by training a range of MLPs with increasing parameter count. See **Section 2.2.1** and **Table D1**.

On the topic of fine-tuning, we maintain the original focus of our work: assessing the **zero-shot information content** of scFM embeddings for perturbation prediction. While fine-tuning is indeed a valuable direction, our primary aim is to **establish a framework for evaluating biologically meaningful perturbation effects in a zero-shot setting**, thereby probing the inherent information content of the embeddings. We therefore believe fine-tuning all the scFMs incorporated into our benchmark is well beyond the scope of this paper. Furthermore, this would require extensive computational and financial resources which we do not currently have at our disposal - and indeed which many people do not have, highlighting the importance of this framework and these results for the research community.

We hope that by incorporating these additional baselines, datasets and experiments, we have managed to address your concerns and that you will take another global look at our submission. We remain committed to advancing this work and we thank you again for your time and your thoughtful feedback.

[1] Replogle, J. M. et al. Mapping information-rich genotype–phenotype landscapes with genome-scale Perturb-seq. Cell 185, 2559–2575 (2022).

---

### Author Response · Authors · 2024-12-02
**Gentle reminder for reviewers - End of discussion period summary**

We want to conclude this rebuttal period by sincerely thanking all the reviewers for their thoughtful feedback and insights. We have successfully obtained further Replogle results. Specifically, we have trained and evaluated most models from split 0.1 to 0.7, including mean and gene expression baselines. The results so far obtained are in line with what we have described in the manuscript, giving further weight to the conclusions reached in the paper. We think that with the inclusion of this dataset, we have addressed the majority of the concerns brought up by the reviewers in their initial reviews, including:

- **MLP parameter count**: Investigation into the effect of model capacity as parameterised by parameter count on our results
- **Dual-gene perturbations**: Inclusion of dual-gene perturbations evaluated across all of the considered settings
- **Statistical significance**: Extension of Figures 3a and 3b to include dual gene perturbations and Spearman $\rho$.
- **Additional Baseline**: Addition of GEARS baseline across single- and dual-gene perturbations in all of the considered evaluation settings
- **Additional Dataset**: Inclusion of the Replogle dataset (most recent checkpoint included below)
- **Clarifications**: We have justified our initial focus and preference for the Norman dataset, highlighting its higher-quality perturbation signal. Furthermore, we have elaborated on the utility of leveraging a zero-shot framework to probe scFM embeddings and have provided a detailed explanation of why we consider fine-tuning is beyond the scope of this work.

We would like to respectfully invite the reviewers to consider if these additions and the extra work carried out during the rebuttal period have sufficiently addressed their concerns. Thank you again for all of your valuable feedback.

**Replogle results checkpoint 02/12/2024**

| Models | Split 0.1 | Split 0.2 | Split 0.3 | Split 0.4 | Split 0.5 | Split 0.6 | Split 0.7 |
|------------------|-----------|-----------|-----------|-----------|-----------|-----------|-----------|
| Mean baseline | 0.21263 | 0.22914 | 0.22276 | 0.21774 | 0.2132 | 0.236 | 0.23873 |
| Raw Baseline | 0.21121 | 0.22697 | 0.22077 | 0.21537 | 0.21198 | 0.23183 | 0.23545|
| scGPT | 0.21145 | 0.22601 | 0.21988 | 0.21542 | 0.21098 | 0.23229 | 0.23334 |
| scFoundation | 0.21075 | 0.226 | 0.21989 | 0.21543 | 0.21102 | 0.26631| 0.23341|
| Geneformer | 0.2107 | 0.226 | 0.21988 | 0.21543 | 0.21098 | 0.24864 | 0.23337 |

---

### Meta-Review · Area_Chair_d6uz · 2024-12-31

**Metareview:**

The paper presents a benchmarking framework for single-cell foundation models on the downstream task of predicting the outcomes of genetic perturbations. The paper has many strengths: the topic is important, the writing is high-quality, and the evaluation is robust in some ways (in particular, it uses a comprehensive set of metrics). However, the use of a single, relatively small dataset was a serious concern for multiple reviewers. Some of the reviewers found the scope of the tasks to be relatively narrow and the absence of a finetuning component to be an issue.

I very much appreciate the authors' responses during the rebuttal period (in particular, the use of the GEARS baseline). However, in the end, we believe it would be better for the authors to consider additional datasets and potentially broaden the paper's scope. Because these additions would be too substantial to pass without an additional round of reviews, I am recommending rejection this time around. We encourage the authors to improve the paper using the feedback in the reviews and to submit it to a different venue.

**Additional Comments On Reviewer Discussion:**

There was a robust discussion between the reviewers and the authors during the rebuttal period. The authors responded to many of the criticisms and provided some additional results. A subset of the scores changed during the discussion.

---

### Decision · Program_Chairs · 2025-01-22

Reject